# Mitochondrial temperature homeostasis resists external metabolic stresses

**Mügen Terzioglu[1]\*, Kristo Veeroja[1], Toni Montonen[1], Teemu O Ihalainen[1], Tiina S Salminen[1], Paule Bénit[2], Pierre Rustin[2], Young-Tae Chang[3], Takeharu Nagai[4], Howard T Jacobs[1,5]\***

[1]Faculty of Medicine and Health Technology, Tampere University, Tampere, Finland; [2]Université Paris Cité, Inserm, Maladies Neurodéveloppementales et Neurovasculaires, Paris, France; [3]SANKEN (The Institute of Scientific and Industrial Research), Osaka University, Ibaraki, Japan; [4]Department of Chemistry, POSTECH, Pohang, Republic of Korea; [5]Department of Environment and Genetics, La Trobe University, Melbourne, Australia

**\*For correspondence:**
mugen.terzi@gmail.com (MT);
howard.jacobs@tuni.fi (HTJ)

**Competing interest:** The authors declare that no competing interests exist.

**Abstract** Based on studies with a fluorescent reporter dye, Mito Thermo Yellow (MTY), and the genetically encoded gTEMP ratiometric fluorescent temperature indicator targeted to mitochondria, the temperature of active mitochondria in four mammalian and one insect cell line was estimated to be up to 15°C above that of the external environment to which the cells were exposed. High mitochondrial temperature was maintained in the face of a variety of metabolic stresses, including substrate starvation or modification, decreased ATP demand due to inhibition of cytosolic protein synthesis, inhibition of the mitochondrial adenine nucleotide transporter and, if an auxiliary pathway for electron transfer was available via the alternative oxidase, even respiratory poisons acting downstream of oxidative phosphorylation (OXPHOS) complex I. We propose that the high temperature of active mitochondria is an inescapable consequence of the biochemistry of OXPHOS and is homeostatically maintained as a primary feature of mitochondrial metabolism.

## eLife assessment

The study provides **useful** data supporting prior findings that mitochondria in cultured cells maintain a temperature up to 15°C above the external temperature at which cultured cells are maintained. The evidence supporting the hypothesis is **solid**, although direct measures of temperature in isolated mitochondria or comparison with other cellular compartments would have strengthened the ability to interpret the relevance of the findings. Nevertheless, the bioenergetic implications of the work will be of interest to cell biologists, biochemists, and physiologists.

## Introduction

The mitochondrial system of oxidative phosphorylation (OXPHOS) is considered the most efficient energy conversion platform of non-photosynthetic eukaryotes. However, it does not operate at or anywhere near 100% thermodynamic efficiency. Much of the free energy released by the reoxidation of primary electron carriers such as NADH is converted to heat. Note that this applies under normal physiological conditions (**Kang, 2018**; **Brand, 2005**; **Nath, 2016**), not only in cells expressing uncoupler proteins or treated with a chemical uncoupler. Furthermore, this heat production is not mere 'waste', since it is a major and regulatable energy source for maintaining body temperature in homeotherms such as mammals and birds (**Brand et al., 1994**), and possibly many other organisms, even those living at a wide range of temperatures in varying environments. Uncoupler proteins merely serve

an auxiliary function under cold stress, as shown by the fact that, in mice, ablation of the major uncoupling protein UCP1 is nonlethal (*Enerbäck et al., 1997*) and does not affect basal body temperature.

Using the mitochondrially targeted fluorescent dye Mito Thermo Yellow (MTY), it was recently shown that intramitochondrial temperature in respiring, human HEK293-derived (immortalised embryonic kidney) cells is at least 10–12°C higher than the ambient temperature at which the cells are maintained (*Chrétien et al., 2018*). In essence, this inference was based on the fact that the MTY fluorescence change, after cells were treated with OXPHOS inhibitors such as rotenone (targeted on OXPHOS complex I, cI), antimycin (targeted on OXPHOS complex III, cIII), or oligomycin (targeted on ATP synthase – OXPHOS complex V, cV), or subjected to prolonged anaerobiosis, indicated a temperature drop of this magnitude, based on internal calibration at the end of each experiment. Inferences from the use of protein-based fluorescence probes targeted to mitochondria are similar (*Nakano et al., 2017*). In addition, similar methods have indicated that the endoplasmic reticulum (ER) (*Kriszt et al., 2017*), cell nucleus (*Okabe et al., 2012*; *Hayashi et al., 2015*), plasma membrane (*Liang et al., 2022*), and different cellular compartments, such as the cell body and neurites of neurons (*Tanimoto et al., 2016*), are at temperatures distinct from the extracellular environment and from each other.

These findings have far-reaching implications for biochemistry and physiology, but need further validation and elaboration before they can be generally accepted. In particular, since the cells of many organisms, notably homeotherms, are unable to tolerate prolonged high or low temperatures when applied externally, the question arises as to how much the temperature of mitochondria is permitted to vary in vivo, and whether and how it is regulated according to physiological conditions (*Lane, 2018*).

In this study we undertook to assess mitochondrial temperature in a number of cultured cell lines from different organisms, and subjected to different metabolic stresses involving well-characterised inhibitors and the transgenic expression of the thermogenic alternative oxidase (AOX) from a marine invertebrate. Broadly, the outcome of these studies verifies that mitochondrial temperature is physiologically maintained at ~15°C above ambient temperature, and is subject to homeostatic regulation in response to metabolic stress.

## Results

### Based on MTY fluorescence mitochondrial temperature is up to ~15°C above ambient

In previous studies (*Chrétien et al., 2018*), MTY fluorescence was tracked in continuously oxygenated human HEK293-derived cells maintained at a constant external temperature, in which mitochondria had been prelabelled with MTY and then treated with one of several OXPHOS inhibitors under controlled oxygen concentration. Internal calibration in each experiment indicated a mitochondrial temperature decrease of 10–12°C (*Chrétien et al., 2018*) following the transition from normoxia to anaerobiosis, or inhibition of the OXPHOS system by classic inhibitors (rotenone, antimycin, cyanide, or oligomycin – see *Figure 1*). We conducted a systematic follow-up study with different inhibitors and cell lines, and sufficient repeats to generate statistical reliability (*Figure 2*), once the MTY fluorescence intensity had reached a stable value following initial oxygenation. See *Figure 2—figure supplement 1*, *Figure 2—figure supplement 2*, *Figure 2—figure supplement 3*, *Figure 2—figure supplement 4*, and *Figure 2—figure supplement 5* for details of the instrumentation, its validation and calibration, and various steps that we undertook to validate the method. These trials showed that the fluorescence signal of MTY in cell-free solution decreases almost linearly with temperature (*Figure 2—figure supplement 3B*), is not responsive to changes in calcium concentration (*Figure 2—figure supplement 4A*), hydrogen peroxide (*Figure 2—figure supplement 4B*), or pH (*Figure 2—figure supplement 4C*), across the physiological range of these parameters. Note that the response curve generated (*Figure 2—figure supplement 3B*), cannot be used directly for calibration in living cells, due to the endogenous contributions of autofluorescence and fluorescence quenching, which are considerable, and will vary according to how much label is actually taken up and retained in mitochondria in each trial. Instead, the internal calibration carried out in each experiment (*Figure 2—figure supplement 3A*) provides a more reliable guide to the temperature changes produced.

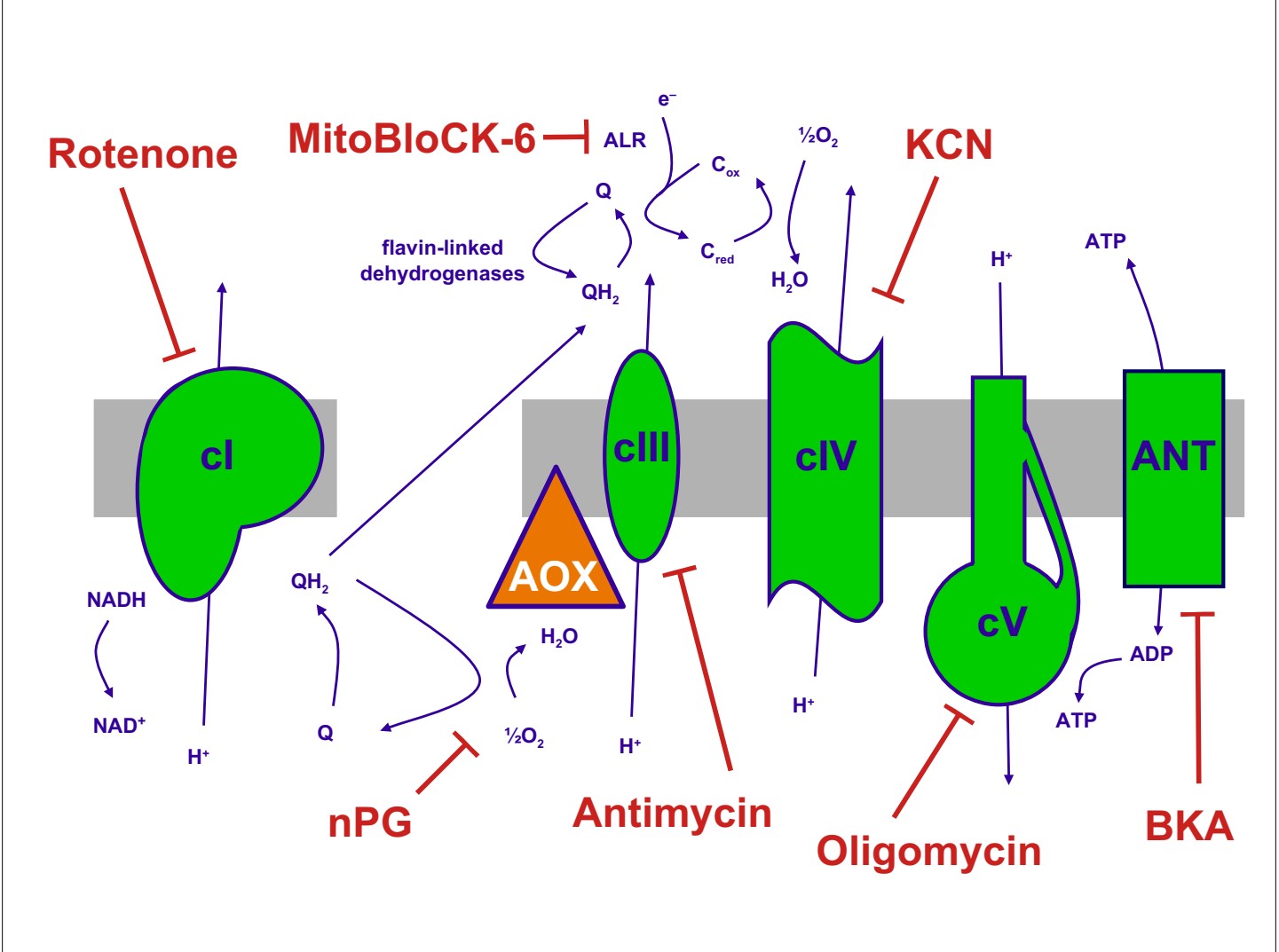

**Figure 1.** The mitochondrial oxidative phosphorylation (OXPHOS) system and inhibitors. Summary of the major components of the OXPHOS system and classic inhibitors. Protonmotive OXPHOS enzyme complexes (cI, cIII, cIV, cV) shown in green, the non-protonmotive transgenically introduced alternative oxidase (AOX) from *Ciona intestinalis* in orange, the inner mitochondrial membrane in grey. Ions and small molecules indicated in purple, inhibitors in brick-red. BKA – bongkrekic acid, an inhibitor of the adenine nucleotide translocase (ANT). nPG – *n*-propyl gallate, an inhibitor of AOX, Q – ubiquinone (oxidised coenzyme Q), $QH_2$ – ubiquinol (reduced coenzyme Q), $c_{red}$, $c_{ox}$ – reduced and oxidised forms of cytochrome *c*, respectively. Complex I (cI), along with a number of flavin-linked dehydrogenases, one of which is succinate dehydrogenase, also described as complex II, each reduce ubiquinone, thus contributing input electrons to complex III (cIII). ALR – 'augmenter of liver regeneration', Evr1 in yeast, acts as an additional feeder of electrons to cytochrome *c*, from the oxidation of sulfhydryl groups in proteins destined for the mitochondrial inter-membrane space via the Mia40 pathway and is inhibited by MitoBLloCK-6.

In most cell lines tested there was a clear gradation of mitochondrial temperature decrease using inhibitors that act progressively further along the OXPHOS chain: the effect was always greatest after oligomycin (inhibitor of cV) and generally least for rotenone (cI inhibitor), with potassium cyanide (inhibitor of OXPHOS complex IV, cIV) and antimycin (cIII inhibitor) intermediate (*Figure 2*). In general, findings were similar for different cell lines, with an implied temperature decrease of the order of 17–19°C produced by oligomycin. This implies a mitochondrial temperature of at least 54°C, assuming that cV inhibition abolishes all mitochondrial heat production, ±2°C, taking account of the possible slight deviation from linearity seen in the MTY fluorescence temperature-response curve at higher temperatures (dotted line, *Figure 2—figure supplement 3B*). Two different isolates of immortalised mouse embryonic fibroblasts (iMEFs) gave similar results (*Figure 2C and D*), with no significant difference in the extent of mitochondrial temperature decrease produced by rotenone to inhibit cI or by

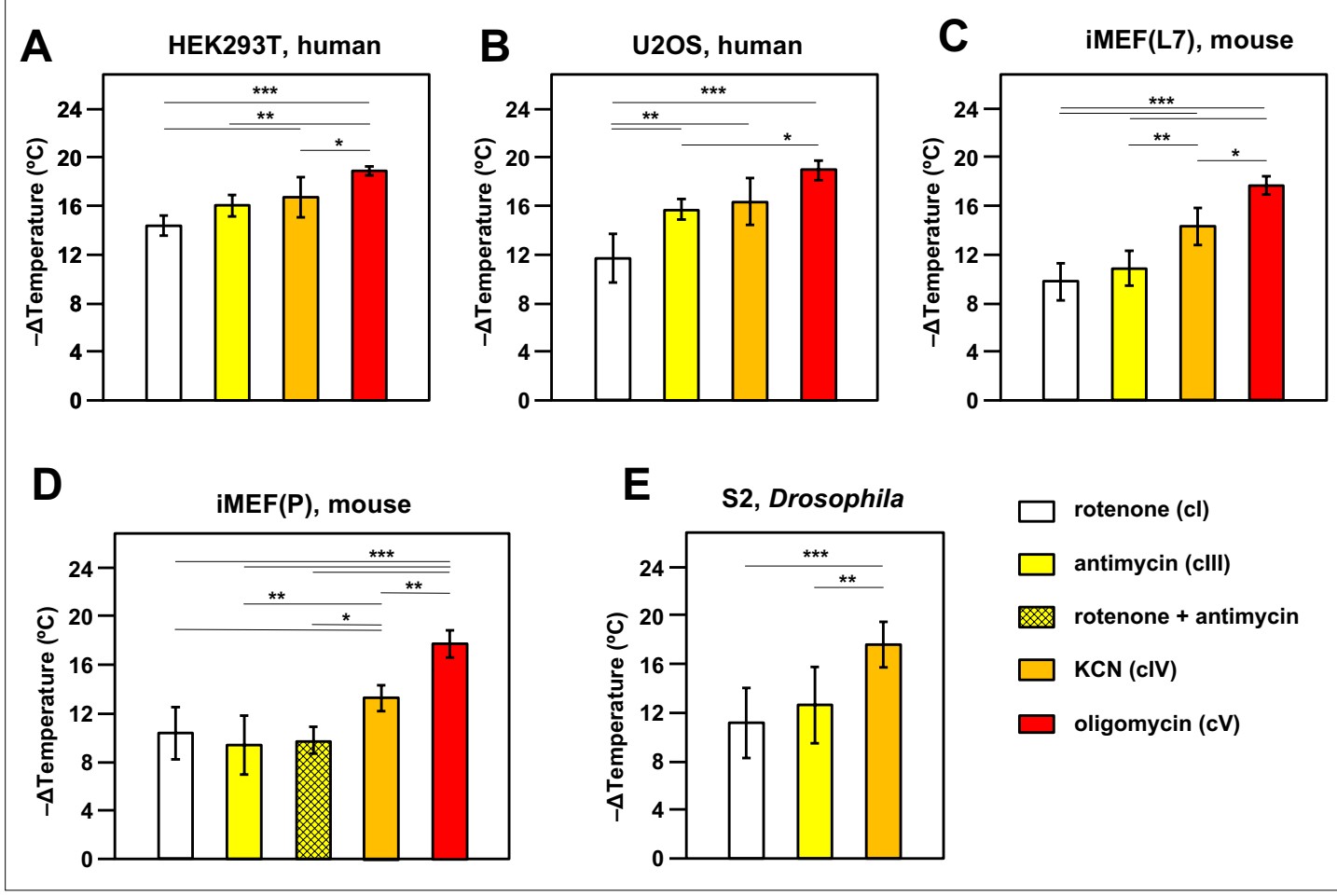

**Figure 2.** Oxidative phosphorylation (OXPHOS) inhibitors decrease mitochondrial temperature to different extents. Extrapolated mitochondrial temperature shifts (means ± SD for at least five independent experiments in each case), based on Mito Thermo Yellow (MTY) fluorescence in (**A-E**) five different cell lines treated with the indicated OXPHOS inhibitors, with internal temperature calibration at the end of each experiment. *, **, *** denote significantly different data sets (one-way ANOVA with Tukey HSD post hoc test, $p < 0.05$, 0.01, and 0.001, respectively). Note that a stable reading could not be obtained for S2 cells treated with oligomycin (*Figure 2—figure supplement 5D*), hence reliable calibration was not possible. iMEF(L7) and iMEF(P) are two isolates of immortalised, embryonic fibroblasts derived from wild-type C57Bl/6J mice.

The online version of this article includes the following source data and figure supplement(s) for figure 2:

**Source data 1.** No title.

**Source data 2.** No title.

**Source data 3.** No title.

**Source data 4.** No title.

**Source data 5.** No title.

**Figure supplement 1.** Instrumentation for mitochondrial temperature estimation using fluorescent probes.

**Figure supplement 2.** Instrument validation steps for estimation of mitochondrial temperature.

**Figure supplement 3.** Fluorescence calibration steps for estimation of mitochondrial temperature.

**Figure supplement 3—source data 1.** No title.

**Figure supplement 4.** Independence of Mito Thermo Yellow (MTY) fluorescence from parameters other than temperature.

**Figure supplement 5.** Supplementary data on effects of oxidative phosphorylation (OXPHOS) inhibitors on mitochondrial temperature.

**Figure supplement 5—source data 1.** No title.

antimycin to inhibit cIII (*Figure 2C and D*) or by a combination of the two drugs (*Figure 2D*). Unlike rotenone plus antimycin (*Figure 2—figure supplement 5A*, panel i) other combinations of OXPHOS inhibitors did not give stable fluorescence values (*Figure 2—figure supplement 5A*, panels ii–vi) and were not considered further. We were also unable to estimate directly the individual contribution to mitochondrial temperature maintenance of the ALR (Evr1) pathway (of mitochondrial protein import to the inter-membrane space [IMS]), in which electrons are fed directly to cytochrome *c* (*Figure 1*): preparations of the ALR inhibitor MitoBloCK-6 (*Dabir et al., 2013*) that we were able to source had a distinct yellow colour that rendered measurements of MTY fluorescence uninterpretable (*Figure 2—figure supplement 5B*).

We investigated the effect of culturing iMEF cells in low-glucose medium, with or without galactose supplementation, aiming to enhance dependence on mitochondrial respiration (*Marroquin et al., 2007*). This culture-medium alteration produced no significant change to the effect of cI inhibition on mitochondrial temperature (*Figure 2—figure supplement 5C*).

The inferred temperature of mitochondria in S2 cells from the fruit-fly *Drosophila*, a poikilotherm, when suspended and oxygenated at 25°C, showed similar responses to OXPHOS inhibition as mammalian cells suspended at 37°C, except that we were unable to derive a stable reading after oligomycin treatment (*Figure 2—figure supplement 5D*, panel i). MTY fluorescence was still increasing 2 hr after the addition of oligomycin, whereas other inhibitors, such as KCN, when applied to S2 cells, gave a stable reading over the same time period (e.g. *Figure 2—figure supplement 5D*, panel ii). Fluorescence reached a stable value after the addition of oligomycin in all of the other cell lines tested, e.g., U2OS human osteosarcoma cells as shown (*Figure 2—figure supplement 5D*, panel iii).

## Protein ratiometric fluorescence confirms mitochondrial temperature measured by MTY

Although the previous study (*Chrétien et al., 2018*) and our own validation steps (*Figure 2—figure supplement 4*) excluded the most obvious possible artefacts from the use of MTY fluorescence as a mitochondrial thermometer, such as an effect of membrane potential, the redox state of the respiratory chain, reactive oxygen species (ROS) production calcium and pH fluxes, artefacts of unknown provenance could always be in play. We therefore set about applying a different method to measure mitochondrial temperature in these cell lines, based on the ratiometric fluorescence of mito-gTEMP (*Nakano et al., 2017*). This technique is based on the fluorescence ratio between two previously developed fluorescent proteins, mT-Sapphire and Sirius, co-expressed as a single, self-cleaved polypeptide, with each cleavage product targeted to mitochondria using a reiterated cytochrome oxidase subunit 8 (COX8) presequence (*Figure 3A*). The basis of the method is the fact that the fluorescence of Sirius is highly temperature sensitive, declining (like MTY) as temperature rises, whereas that of mT-Sapphire is not. We stably transfected HEK293T, U2OS, and iMEF(P) cells with mito-gTEMP and confirmed mitochondrial targeting, as exemplified in *Figure 3B* and *Figure 3—figure supplement 1*. Since the two proteins are expressed at a fixed stoichiometry, the peak fluorescence ratio is independent of expression level, which may vary even within a cloned mammalian cell population (*Pilbrough et al., 2009*), as seen, for example, in *Figure 3B*.

To control for possible artefacts introduced by mito-gTEMP in the presence of OXPHOS inhibitors, we tested whether the drugs alone in PBS affect background fluorescence in the absence of cells. Background fluorescence at the wavelengths used to track mT-Sapphire and Sirius was both more than an order of magnitude less than that of the fluorescent proteins expressed in iMEFs, and was unaffected by rotenone (*Figure 3—figure supplement 2A*). Background fluorescence at both wavelengths was slightly increased by oligomycin (*Figure 3—figure supplement 2B*), but the fluorescence ratio was unchanged (*Figure 3—figure supplement 2C*). In contrast, antimycin produced a substantial change in background fluorescence, especially at the Sirius wavelength (*Figure 3—figure supplement 2D*), which affected the fluorescence ratio (*Figure 3—figure supplement 2E*), thus making it unsuitable for use in estimating mitochondrial temperature shifts.

Since, based on MTY fluorescence, we had inferred a maximal amount of mitochondrial temperature decrease in the presence of oligomycin, we used oligomycin-treated cells to calibrate the mito-gTEMP fluorescence ratio against externally imposed temperature. This assumes that any residual mitochondrial heat production in oligomycin-treated cells is negligible and invariant with respect to externally applied temperature changes.

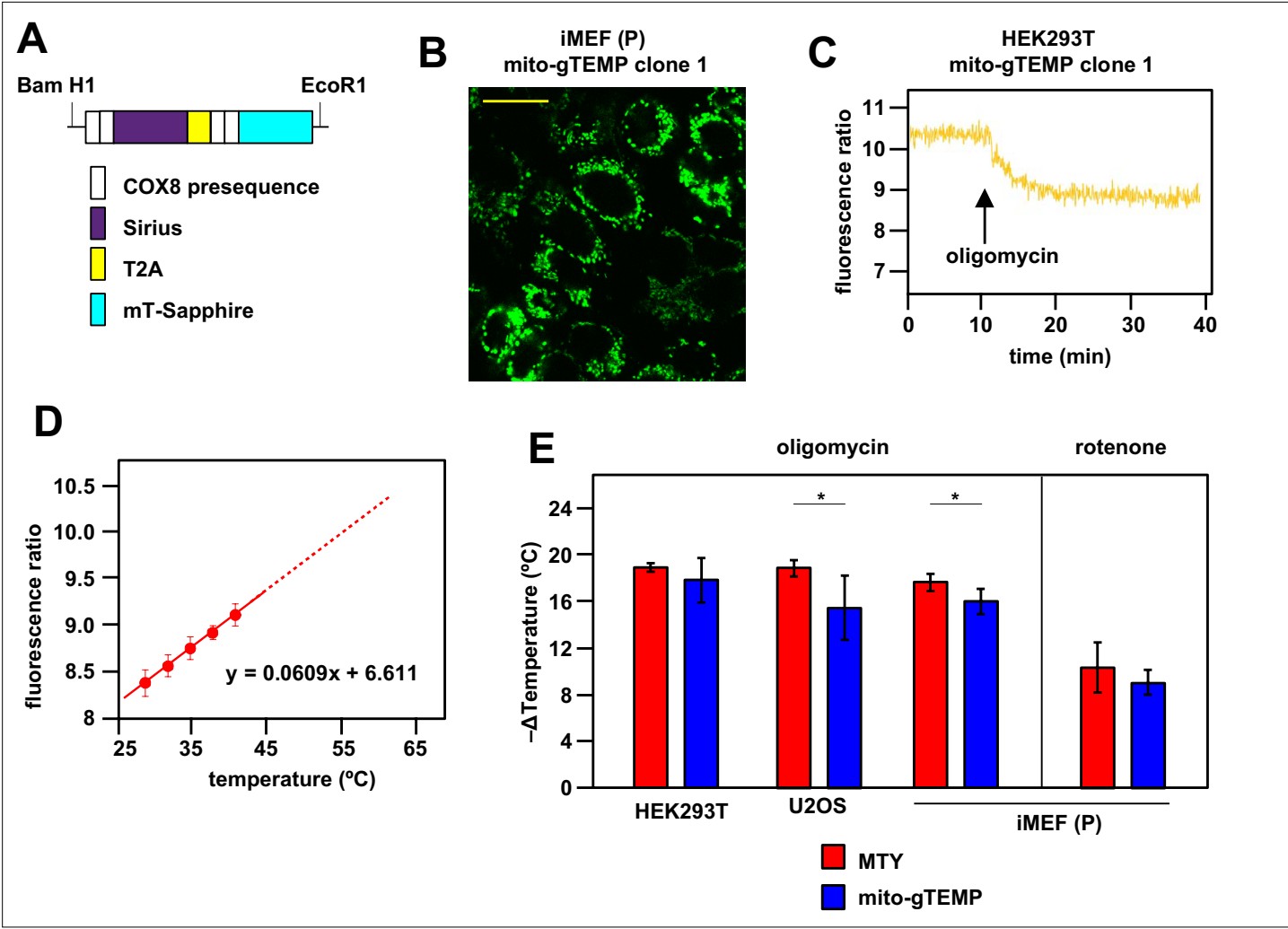

**Figure 3.** Ratiometric fluorescence probes confirm mitochondrial temperature measurements by Mito Thermo Yellow (MTY). (**A**) Diagrammatic map (not to scale) of the self-cleaving polyprotein encoded by plasmid mito-gTEMP (coxVIIIx2-gTEMP)/pcDNA3, indicating the reiterated cytochrome oxidase subunit 8 (COX8) targeting presequences, the Sirius and mT-Sapphire coding regions and the T2A element (*Donnelly et al., 2001a*), which promotes self-cleavage via ribosome frameshifting (*Donnelly et al., 2001b*). (**B**) Illustrative mT-Sapphire fluorescence image of iMEF cell clone expressing mito-gTEMP, showing variable expression even within cells of the clone. Scale bar 5 µm. See *Figure 3—figure supplement 1* for confirmation of mitochondrial localisation. (**C**) Illustrative trace of mT-Sapphire/Sirius fluorescence ratio in HEK293T cell clone expressing mito-gTEMP, following treatment with oligomycin. Note that, as for MTY, Sirius fluorescence decreases with increasing temperature, so that the decreased mT-Sapphire/Sirius fluorescence ratio after oligomycin treatment represents a decrease in temperature. (**D**) Calibration curve for mT-Sapphire/Sirius fluorescence ratio (means ± SD, nine independent experiments in each case) in mito-gTEMP expressing iMEFs treated with oligomycin, and shifted to various temperatures up to 42°C. Solid red line represents linear best fit (as per equation shown) extrapolated to higher temperatures (dotted red line) that were not directly tested due to the effects of prolonged high temperatures on cell integrity and viability (*Roti Roti, 2008*). Note that, during these trials, we observed that continuous illumination of mito-gTEMP expressing cells at the excitation wavelength for more than 40 min caused damage affecting the fluorescence signal. All calibration measurements were therefore conducted over <35 min. We also noted that the internal temperature of the cuvette took ~2 min to equilibrate with that of the surrounding Peltier (*Figure 2—figure supplement 2*), for which reason fluorescence values were measured over just the final minute at each calibration step. (**E**) Extrapolated mitochondrial temperature shifts (means ± SD for at least five independent experiments in each case) in the indicated cell lines and mito-gTEMP transfected cell clones following treatment with oligomycin, based on (red bars) MTY fluorescence, reproduced from corresponding panels of *Figure 2*, and (blue bars) mito-gTEMP fluorescence ratio, applying calibration curves such as shown in panel (**C**), derived separately for each cell line studied. ** denotes significant differences between the two methods (Student's t test within each cell line, p<0.01).

The online version of this article includes the following source data and figure supplement(s) for figure 3:

**Source data 1.** No title.

**Source data 2.** No title.

*Figure 3 continued on next page*

*Figure 3 continued*

**Figure supplement 1.** Mitochondrial localisation of mito-gTEMP.

**Figure supplement 2.** Supplementary data on mitochondrially targeted temperature-sensitive fluorescent probes.

**Figure supplement 3.** Idealised scheme for cellular/mitochondrial fractionation.

After oligomycin treatment, iMEF mitochondria showed a decline in the mito-gTEMP fluorescence ratio over ~10 min, thereafter holding a constant value for at least a further 25 min that is assumed to represent the ambient temperature (*Figure 3C*). This enabled us to construct an in vivo calibration curve for the mito-gTEMP fluorescence ratio against externally imposed temperature (*Figure 3D*), which fitted a linear function over the temperature range at which we were able to conduct the measurement: higher temperatures result in loss of cell integrity and viability (*Roti Roti, 2008*), so were not used. Extrapolation to higher mitochondrial temperatures is therefore based on the assumption that the relationship remains linear, as was determined previously for the isolated gTEMP proteins (*Nakano et al., 2017*).

Oligomycin treatment of mito-gTEMP-expressing cell clones derived from three different mammalian cell lines gave extrapolated values for the mitochondrial temperature decrease in the same range as those inferred by MTY, although slightly (1–2°C) less in each case, implying an internal mitochondrial matrix temperature of ~52–54°C (*Figure 3E*). Rotenone treatment of iMEF(P) cells also produced a similar amount of mitochondrial temperature decrease when estimated by mito-gTEMP as by MTY (*Figure 3E*).

The slight discrepancy in the mitochondrial temperature inferred by the two methods prompted us to consider whether mito-gTEMP and MTY are targeted to different sub-mitochondrial compartments. The matrix localisation of GFP-related fluorescent reporters using the COX8 targeting peptide is well documented (*Molina and Shirihai, 2009*). However, the exact intramitochondrial location of MTY is not known, besides the fact that at least some of it interacts with the matrix protein aldehyde dehydrogenase 2 (ALDH2) (*Arai et al., 2015*). To address this issue experimentally, we subfractionated mitochondria from cells pre-stained with MTY (see *Figure 3—figure supplement 3*) and measured fluorescence at 562 nm, subtracting the autofluorescence manifested by the corresponding fraction from unstained cells prepared in parallel. To prevent dye leakage, we supplied a standard pyruvate-glutamate-malate substrate mix to maintain mitochondria in the energised state throughout the procedure. In the first fractionation step, mitoplasts were separated from the outer membrane space/IMS fraction. In two trials, the mean proportion of MTY fluorescence retained in mitoplasts was 92%. In the second fractionation step, mitoplasts were sonicated to prepare fractions enriched for matrix versus 'inside-out' sub-mitochondrial particles (SMPs). In two trials, the mean proportion of MTY fluorescence retained in the SMP fraction was 90%. These findings are consistent with MTY being retained within, or closely associated with, the inner mitochondrial membrane.

## Individual cells and organelles show mitochondrial temperature differences and fluctuations

As studied by either method, mitochondrial temperature represents an average over many cells, each containing many mitochondria. We used confocal imagining to look at the profile of mitochondrial temperature in individual iMEF(P) cells and organelles stained with MTY (*Figure 4A*, panel i). This revealed much greater heterogeneity between and even within cells not seen, for example, with the ubiquitous mitochondrial stain MitoTracker Deep Red FM (*Figure 4A*, panel ii). Within individual cells stained with MTY we observed some more brightly staining puncta (*Figure 4B*, *Video 1*). Time-lapse imaging revealed that mitochondria in individual cells were cooled to variable extents by oligomycin treatment (*Figure 4C*).

## Metabolic adaptation can maintain or restore mitochondrial temperature

When mito-gTEMP-expressing iMEF cells were incubated in PBS at 37°C without the addition of any inhibitor, we observed that, after reaching a steady fluorescence ratio indicative of a temperature of 53–54°C, there was a modest but progressive decline in the fluorescence ratio, such that by 35 min of incubation the inferred mitochondrial temperature was around 4°C lower than the peak

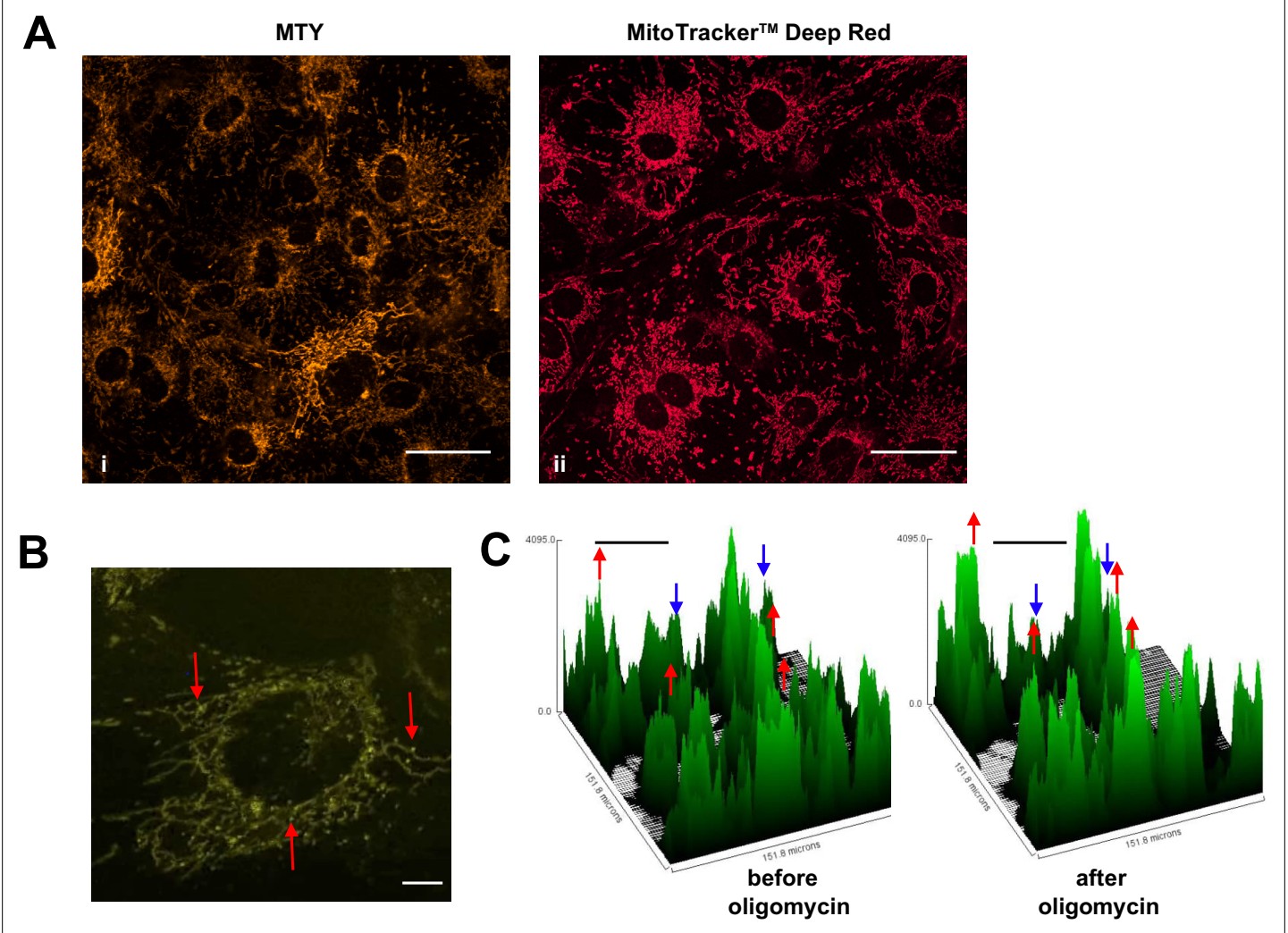

**Figure 4.** Mitochondrial temperature variation between and within cells. (**A, B**) Micrographic images of iMEF cells stained with Mito Thermo Yellow (MTY) (A, panel i and B), alongside a parallel culture of cells stained with Mitotracker Deep Red FM (A, panel ii), which shows much more uniform staining. The intensity of MTY staining appears to vary between and even within cells. Scale bars – 10 μm. (**B**) Still image from *Video 1*, brightness and contrast optimised, showing bright (i.e. potentially cooler) foci (arrowed) within the less intensely stained mitochondrial network. Many of these foci appear mobile within the mitochondrial network, over the 5 min time-scale of the video. Scale bar – 2 μm. (**C**) Fluorescence intensity histograms of a field of cells immediately before and 33 min after oligomycin treatment. Note that, in addition to the general pattern of increased brightness (e.g. peaks represented by red dots), mitochondria in some cells showed only minor changes in fluorescence intensity (e.g. peaks denoted by blue dots). Scale bars – 50 μm.

value reached at approximately 10 min (*Figure 5A and B*). Although remaining much warmer than the surrounding medium, this suggests that, in the complete absence of an externally supplied, metabolisable substrate, mitochondrial metabolism adjusts to a slightly lower heat output and steady-state temperature. To explore this phenomenon further we added the main substrates normally provided to mammalian cells in culture, namely glucose and glutamine. As expected, this enabled mitochondria to maintain a higher temperature of ~53°C at the end of the experiment, although this was still slightly less than the initial temperature peak of ~55°C between 5 and 10 min of incubation (*Figure 5B and C*). However, the addition of glucose and glutamine at 10 min did produce a small rise in mitochondrial temperature (*Figure 5B and D*). In sum, these data indicate that mitochondrial temperature fluctuates within a narrow range according to substrate supply.

In order to explore the issue further, we treated cells with bongkrekic acid (BKA), a potent inhibitor of the mitochondrial adenine nucleotide translocase (ANT) (*Henderson and Lardy, 1970*). The ANT exchanges ADP and ATP across the mitochondrial inner membrane (IMM), thus enabling OXPHOS to

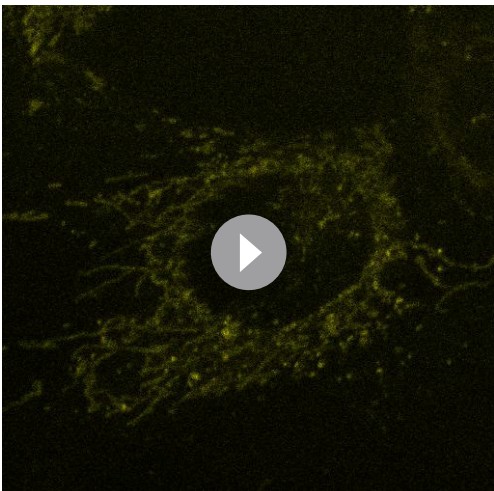

**Video 1.** Mitochondrial temperature variation between and within cells. Time-lapse micrographic images (1 frame per 10 s) of iMEF cells stained with Mito Thermo Yellow (MTY), brightness and contrast optimised, showing bright (i.e., potentially cooler) foci within the less intensely stained mitochondrial network. Many of these foci appear mobile within the mitochondrial network, over the 5 min time-scale of the video. See *Figure 4B* for a still image taken from the video.
https://elifesciences.org/articles/89232/figures#video1

supply the rest of the cell with ATP. Previous investigators have shown that cultured mouse fibroblasts respond to BKA by inducing glycolysis and downregulating mitochondrial respiration, whilst maintaining a high mitochondrial membrane potential (*Kano et al., 2019*). In initial trials we found that BKA quenches the fluorescence of MTY, even in the absence of cells (*Figure 5—figure supplement 1*). Therefore, we used the mito-gTEMP fluorescence ratio, which was less disturbed by the drug, to estimate its effects on mitochondrial temperature. We reasoned that the metabolic switching induced by BKA should result in a substantial mitochondrial temperature decrease, due to the expected lowering of flux through the respiratory chain. However, the mitochondrial temperature decrease that we observed was minimal, following an apparent slight increase upon addition of the drug (*Figure 5E and F*), qualitatively consistent with the MTY signal after discounting the quenching effect (*Figure 5—figure supplement 1*).

## AOX maintains or restores high mitochondrial temperature in the presence of OXPHOS inhibitors

Next we looked at iMEF cells constitutively expressing the non-protonmotive AOX from *C. intestinalis*, which provides an alternative pathway for respiratory electron flow from ubiquinol to oxygen, if cIII and/or cIV are inhibited (*Dassa et al., 2009*). We first verified by respirometry that iMEF AOX cells were resistant to inhibitors of cIII (antimycin, *Figure 6—figure supplement 1A*, ) and cIV (KCN, *Figure 6—figure supplement 1B*).

In cells that principally metabolise cI-linked substrates, AOX makes only a minor contribution to respiratory electron flow under non-inhibited conditions (*Szibor et al., 2020*). However, if cI is blocked, forcing cells to switch to other ubiquinone-linked substrates, AOX becomes engaged (*Szibor et al., 2020*). Once again, the cell must switch partially to glycolysis to sustain ATP levels, and to cytosolic NADH reoxidation via lactate dehydrogenase.

Consistent with these expectations, the initial mitochondrial temperature decrease produced by the cI inhibitor rotenone in AOX-expressing iMEFs was very similar to that of control iMEFs, with a mitochondrial temperature decrease of ~10°C based on both MTY fluorescence (*Figure 6A*, panel i, *Figure 6B*) and the mito-gTEMP fluorescence ratio measured in iMEF cells stably transfected with AOX (*Figure 6B*, *Figure 6—figure supplement 2A*). However, this was followed by a gradual mitochondrial rewarming (*Figure 6A*, panel i, *Figure 6C*, *Figure 6—figure supplement 2A*, panel ii). AOX-expressing iMEFs treated with other OXPHOS inhibitors showed a similar behaviour (*Figure 6A* panels ii–iv, *Figure 6C*), except that the initial mitochondrial temperature decrease was substantially less than that seen in control iMEFs (*Figure 6A* panels ii–iv, *Figure 6B*), whilst the rate and extent of rewarming differed according to the inhibitor used (*Figure 6C*). We were not able to estimate directly the contribution of AOX to temperature maintenance, due to fluorescence quenching by the AOX inhibitor *n*-propyl gallate (nPG) (*Figure 6—figure supplement 2B*).

## Mitochondrial temperature is maintained despite varying ATP demand

To model the effects on mitochondrial temperature of a more physiologically relevant perturbation, we downregulated cytosolic protein synthesis, the major cellular consumer of ATP, by treating iMEFs with anisomycin (*Figure 7*). Using our earlier calibrations, we initially estimated the mitochondrial temperature in cells prior to anisomycin treatment, then followed the effect of anisomycin treatment

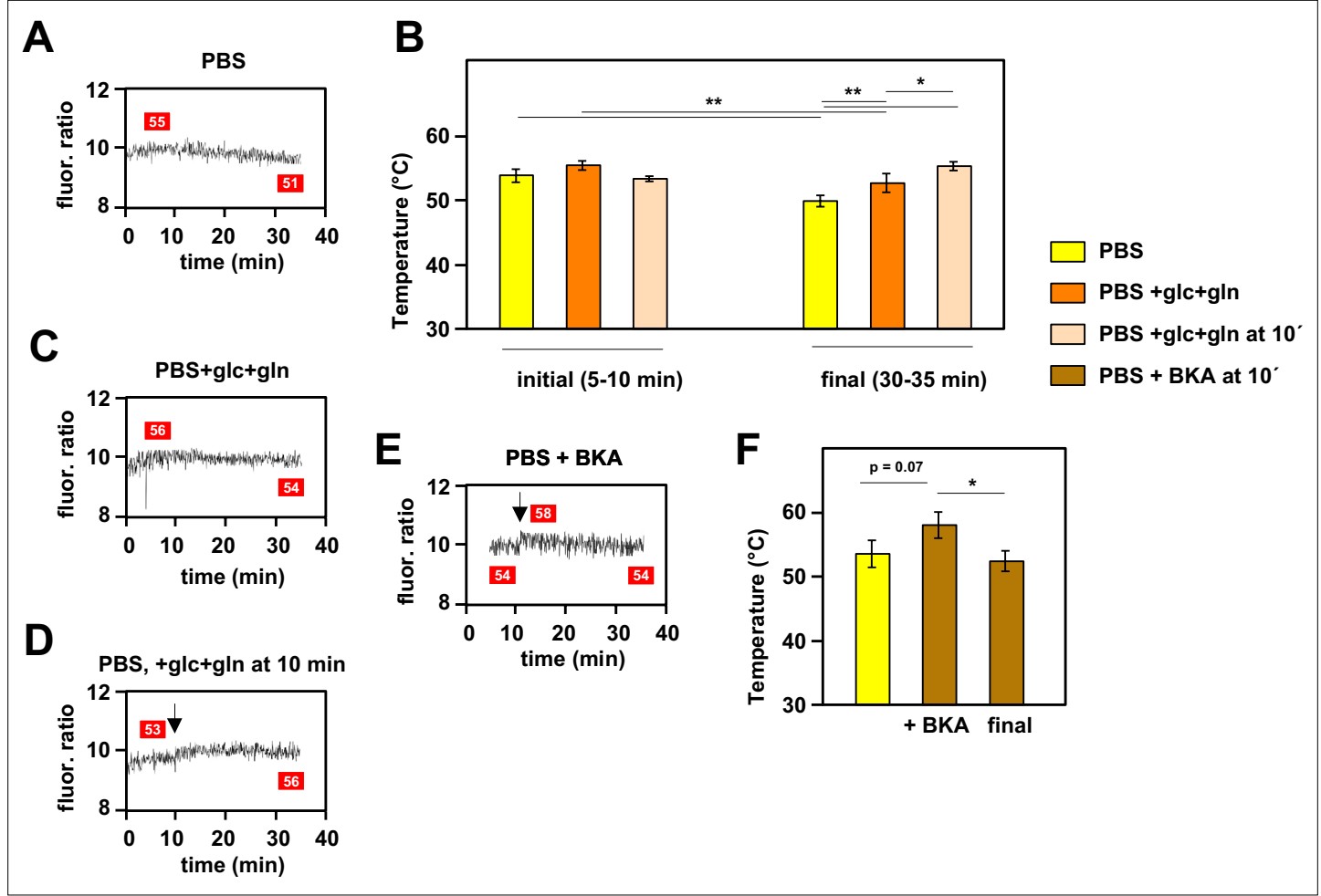

**Figure 5.** Mitochondrial temperature fluctuates in response to substrate availability. (**A, C, D, E**) Representative fluorescence ratio traces of mito-gTEMP-expressing iMEF(P) cells resuspended in various media without oxidative phosphorylation (OXPHOS) inhibitors. PBS – phosphate-buffered saline, glc – glucose, and gln – glutamine at standard cell-culture concentrations, BKA – bongkrekic acid. Arrows indicate time of addition of substrates or of BKA. Mitochondrial temperatures (°C) extrapolated from calibration curve (**Figure 3D**) shown in red boxes. (**B**) Mitochondrial temperatures (°C, means ± SD of five independent experiments), averaged over the time intervals since cell resuspension as shown, extrapolated from mito-gTEMP fluorescence ratios using calibration curve (**Figure 3D**), for cells with and without substrate additions as shown. *, **, *** denote statistical significance (p<0.05, 0.01, 0.001, respectively, one-way ANOVA with Tukey post hoc HSD test). (**F**) Mitochondrial temperatures (°C, means ± SD, n=3 independent experiments), averaged over the time intervals since cell resuspension as shown, extrapolated from mito-gTEMP fluorescence ratios using calibration curve (**Figure 3D**), for cells treated with BKA at 11 min (values averaged over 11–16 min) and then tracked until 35 min (final values averaged over 30–35 min). *, **, *** denote statistical significance (p<0.05, 0.01, 0.001, respectively, one-way ANOVA with Tukey post hoc HSD test).

The online version of this article includes the following source data and figure supplement(s) for figure 5:

**Source data 1.** No title.

**Source data 2.** No title.

**Figure supplement 1.** Effect of bongkrekic acid (BKA) on Mito Thermo Yellow (MTY) fluorescence.

for up to 18 hr (**Figure 7**). In contrast to our expectation that anisomycin, by attenuating ATP demand, would decrease electron flux in the OXPHOS system and thus lead to a drop in mitochondrial temperature, the latter instead showed a transient slight increase (**Figure 7A**). Thereafter, high mitochondrial temperature was sustained for many hours, although by 18 hr of exposure to the drug many of the cells were no longer attached and probably dying (the cells analysed in **Figure 7A** were those that had remained attached). AOX-expressing iMEF cells (**Figure 7B**) showed an almost identical behaviour, except that the cells remained healthy-looking even after 18 hr in anisomycin. To evaluate the contribution of cI and cV to the maintenance of mitochondrial temperature in anisomycin-treated cells, we stained anisomycin-pretreated iMEFs with MTY, then tracked fluorescence changes after treatment

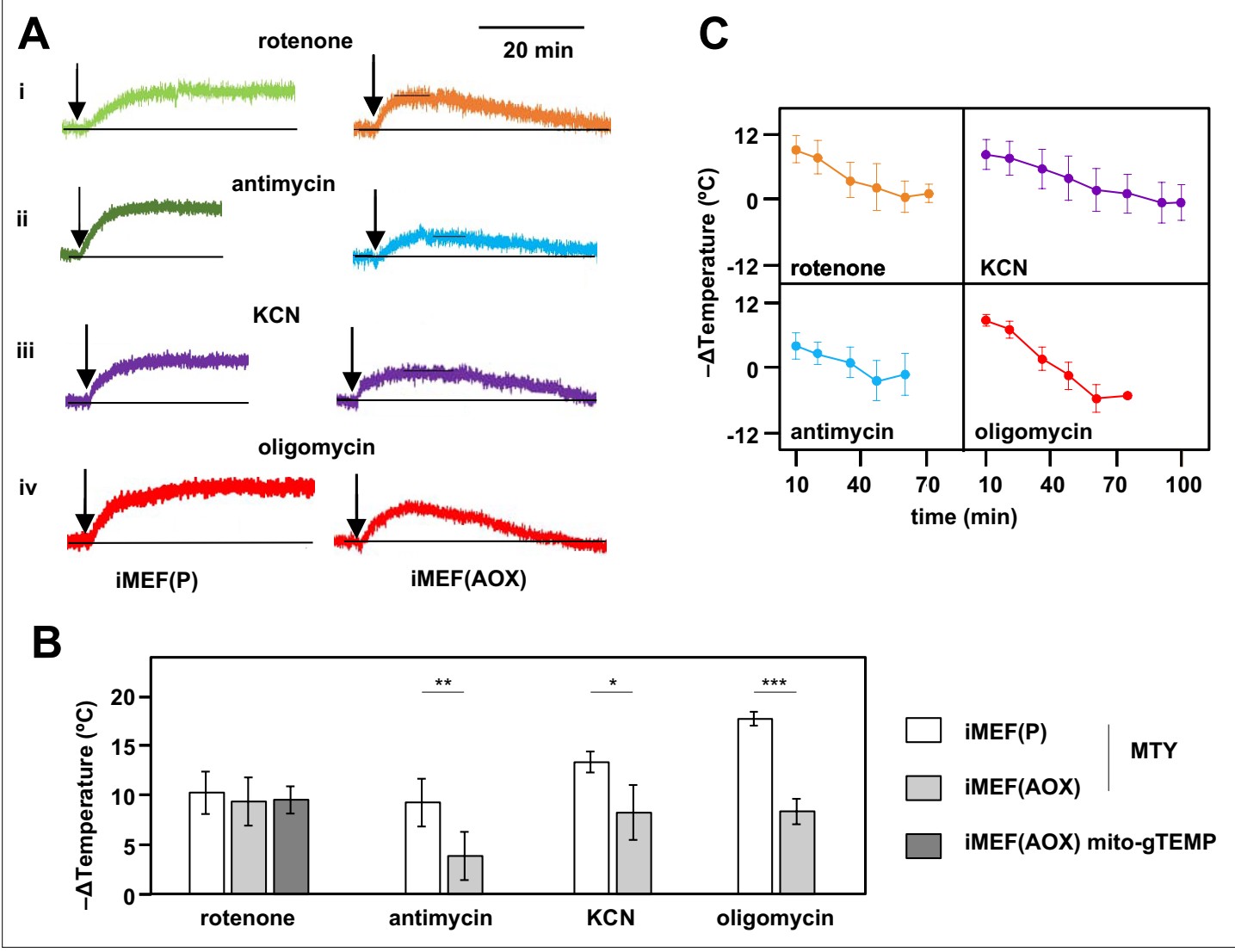

**Figure 6.** Restoration of high mitochondrial temperature in alternative oxidase (AOX)-expressing cells after oxidative phosphorylation (OXPHOS) inhibition. (**A**) Representative traces of Mito Thermo Yellow (MTY) fluorescence over time, in control and AOX-expressing iMEFs treated with the OXPHOS inhibitors shown. Vertical axes are arbitrary values, horizontal axes scaled as indicated. Note that increased fluorescence indicates cooler mitochondria. Control iMEF(P) cells reached and maintained plateau levels of fluorescence, indicating a sustained decrease in mitochondrial temperature, whilst iMEF(AOX) cells showed rewarming after reaching a peak fluorescence value. (**B**) Peak mitochondrial temperature decrease (means ± SD, n>4 independent experiments) in the indicated cell lines following treatment with OXPHOS inhibitors, inferred from MTY fluorescence or mito-gTEMP fluorescence ratio as shown. Horizontal bars indicate significant differences for pairwise comparisons between iMEF(P) and iMEF(AOX) cells treated with the indicated inhibitors (MTY data, Student's t test, p<0.05, 0.01, and 0.001, denoted, respectively, by *, **, and ***). There was no significant difference between the methods, applied to rotenone-treated iMEF(AOX) cells (Student's t test, p>0.05). (**C**) Mitochondrial rewarming (means ± SD, n>5 independent experiments except at latest time-points, where n=3 or 4) in iMEF(AOX) cells treated with the inhibitors shown, inferred from MTY fluorescence traces.

The online version of this article includes the following source data and figure supplement(s) for figure 6:

**Source data 1.** No title.

**Source data 2.** No title.

**Figure supplement 1.** Alternative oxidase (AOX)-expressing iMEFs show antimycin- and KCN-resistant respiration.

**Figure supplement 2.** Supplementary data on effects of inhibitors on fluorescence in alternative oxidase (AOX)-expressing cells.

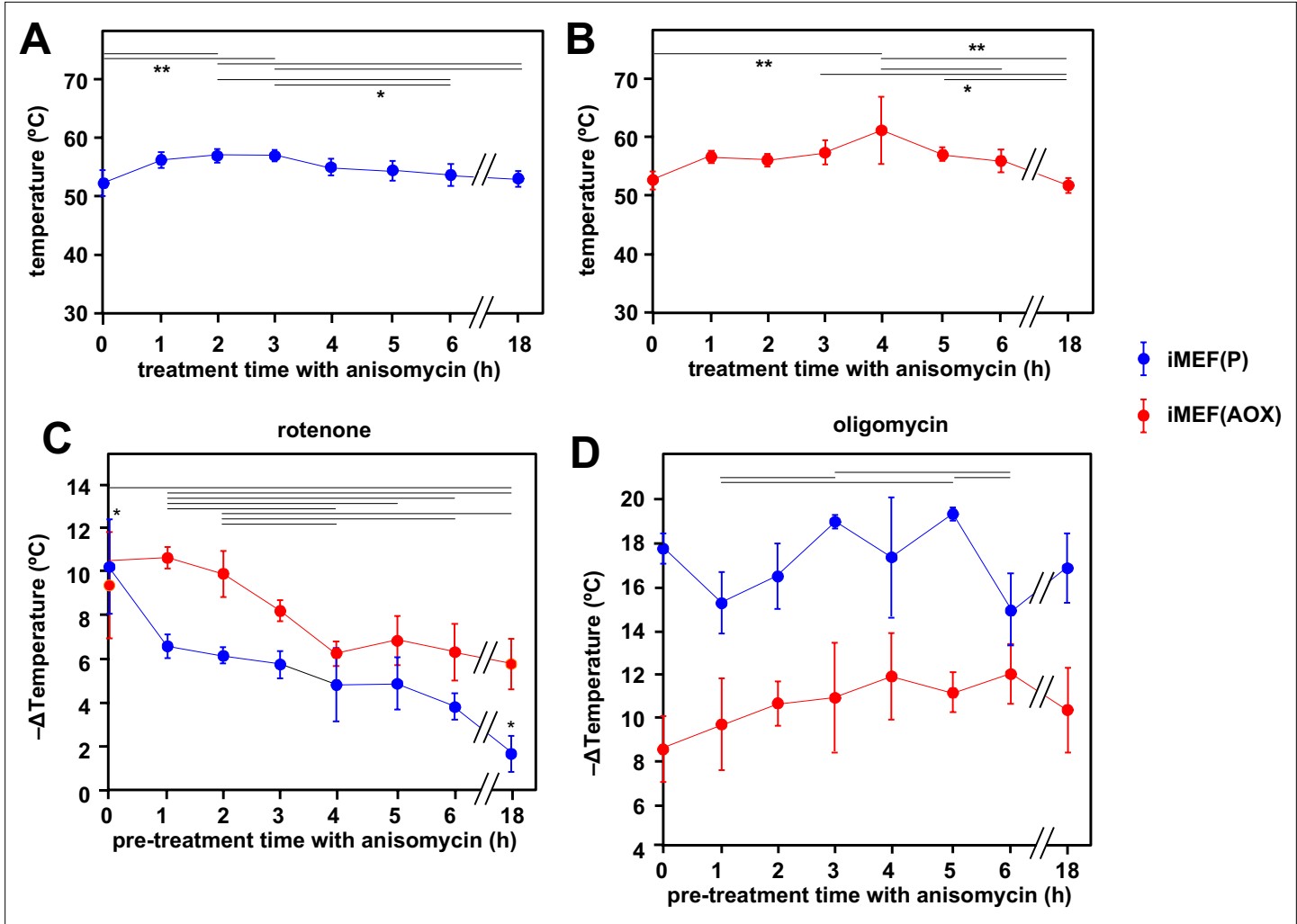

**Figure 7.** Cells maintain high mitochondrial temperature despite varying ATP demand. (**A, B**) Intramitochondrial temperatures (means ± SD) estimated by mito-gTEMP fluorescence ratio in iMEF(P) and iMEF(alternative oxidase [AOX]) cells as indicated, treated with 150 μM anisomycin for the times shown. Temperatures extrapolated from calibration curve shown in *Figure 3D*. Horizontal bars denote data points that were significantly different (one-way ANOVA with Tukey HSD test, p<0.05). (**C**) Extrapolated mitochondrial temperature shifts (means ± SD for at least four independent experiments in each case) based on Mito Thermo Yellow (MTY) fluorescence, as indicated, in iMEF(P) and iMEF(AOX) cells pre-treated with 150 μM anisomycin for the times shown, followed by treatment with rotenone. Zero-time values (i.e. cells not treated with anisomycin) are reproduced from *Figures 2D and 5B* for iMEF(P) and iMEF(AOX) cells, respectively. Values for the two cell lines are significantly different at each time-point except 0 and 4 hr (Student's t test, p<0.05). Within each cell line, in addition to the clear downward trend with time (i.e. lesser amount of mitochondrial temperature decrease), the values for the extreme time-points (0 and 18 hr) are significantly different from each other and from all other time-points for the iMEF(P) line (denoted by asterisks), whilst for the iMEF(AOX) line significant pairwise differences are indicated by horizontal lines above the graphed values (one-way ANOVA with Tukey post hoc HSD test, p>0.05). (**D**) Extrapolated mitochondrial temperature shifts (means ± SD for at least four independent experiments in each case) based on MTY fluorescence, as indicated, in iMEF(P) and iMEF(AOX) cells pre-treated with 150 μM anisomycin for the times shown, followed by treatment with oligomycin. Zero-time values (i.e. cells not treated with anisomycin) are reproduced from *Figures 2D and 5B* for iMEF(P) and iMEF(AOX) cells, respectively. Values for the two cell lines are significantly different at each time-point (Student's t test, p<0.05). Within each cell line, and despite the apparent trend of the means for iMEF(AOX) cells, the values for different time-points are, in general, not significantly different from each other (one-way ANOVA with Tukey post hoc HSD test, p>0.05). Exceptions for iMEF(P) cells are indicated by horizontal lines (p<0.01).

The online version of this article includes the following source data and figure supplement(s) for figure 7:

**Source data 1.** No title.

**Source data 2.** No title.

**Source data 3.** No title.

**Source data 4.** No title.

**Figure supplement 1.** Supplementary data on mitochondrial temperature decrease in anisomycin-treated cells.

with rotenone (*Figure 7C*) or oligomycin (*Figure 7D*). The inferred mitochondrial temperature of anisomycin-treated iMEFs showed a lesser effect of rotenone than iMEFs not treated with anisomycin (*Figure 7C*, blue trace, compare 1–18 hr time-points with 0 hr control), and this effect was increasingly pronounced after longer times of anisomycin exposure. Mitochondrial temperature in the surviving cells after 18 hr in anisomycin was almost unaffected by rotenone. In contrast, and despite some experiment-to-experiment variation, anisomycin-exposed iMEFs were as affected by oligomycin as prior to treatment, even up to 18 hr (*Figure 7D*, blue trace). AOX-expressing iMEFs behaved differently. After anisomycin treatment, they showed a consistently greater mitochondrial temperature decrease produced by rotenone (*Figure 7C*, red trace) than did control iMEFs not expressing AOX (*Figure 7C*, blue trace). Although the effect of rotenone declined with increasing times of anisomycin exposure, it did so more gradually than for control iMEFs not expressing AOX (*Figure 7C*). The mitochondria of anisomycin-treated iMEF(AOX) cells showed a temperature decrease by oligomycin significantly less than anisomycin-treated control iMEF(P) cells, regardless of the length of time of anisomycin exposure, but also showed a possibly increased effect of oligomycin, with increasing anisomycin exposure time (*Figure 7D*, red trace), at least for the first hours. Anisomycin-exposed iMEF(AOX) cells showed the same slow rewarming after rotenone treatment (*Figure 7—figure supplement 1A*, panel i) as did iMEF(AOX) cells not exposed to anisomycin (*Figure 6A* panel i, *Figure 6C*), whilst control iMEFs did not show any such effect from rotenone (*Figure 7—figure supplement 1A*, panel ii). Conversely, the rewarming effect after oligomycin treatment was slower and of much lower magnitude for anisomycin-treated than untreated iMEF(AOX) cells (*Figure 7—figure supplement 1B*), whilst a modest rewarming was also seen for control iMEFs not expressing AOX (*Figure 7—figure supplement 1B*).

## Discussion

In this study, using the mitochondrially targeted, temperature-sensitive dye MTY applied to five cell lines, we confirmed that physiologically active mitochondria in vivo exhibit high temperature, at least 15°C above that of the external environment, i.e., ~52°C in mammalian cells. We validated the measurements using the mito-gTEMP system, based on the ratiometric fluorescence of two mito-chondrially localised reporter proteins. We visualised mitochondrial temperature fluctuations at the single-cell level, and documented the maintenance of high mitochondrial temperature despite variations in growth conditions, substrate availability, ATP demand, and even OXPHOS poisons where an alternative respiratory pathway was available.

No chemical reaction is 100% thermally efficient, and those of OXPHOS are no exception, with up to 60% of the energy yield from the biological oxidations of respiration and ATP synthesis released as heat (*Brand, 2005*; *Nath, 2016*). It has long been assumed that this heat was simply radiated to the rest of the cell or the external environment, maintaining intracellular temperature at a uniform 37–42°C in mammals and birds, or at a variable, environmentally determined temperature in poiki-lotherms such as insects, generally considered as ectothermic. Our findings are inconsistent with this model, indicating instead that much of the heat generated by OXPHOS is retained within the organelle, keeping mitochondrial temperature far above that of other cell components and, whenever possible, within a narrow range. Heat conductance from the mitochondria might also be modulated to maintain mitochondrial temperature.

## Validation and calibration of high mitochondrial temperature

In previous studies using MTY in human HEK293 and HeLa cells, we documented fluorescence changes after treatment with OXPHOS inhibitors that were indicative of high mitochondrial temperature (*Chré-tien et al., 2018*; *Chrétien et al., 2020*). In the present study we extended these findings to other cell lines, including one from *Drosophila*, and two isolates of iMEFs, immortalised using a different cocktail of factors (*Andjelković et al., 2018*; *Giordano et al., 2019*). We systematically conducted multiple repeats with each of five drugs impacting the OXPHOS system. In each case, we implemented an internal calibration step after MTY fluorescence had reached a plateau level in the presence of the inhibitor, by abruptly shifting the external temperature up, then down by 3°C or 5°C, at minimally 4 min intervals. In each case, it is assumed that the observed fluorescence shift is due only to the change in the externally applied temperature (i.e. that any 'corrective' response of the cell in the presence of the inhibitor is negligible). This is supported by the fact that the new fluorescence values reached

during the calibration steps are essentially constant over the time-scales used (see *Figure 2—figure supplement 3A* for examples). Internal calibration in each experiment provides a control for variation in the exact number of cells or the amount of label taken up. The use of a Peltier jacket surrounding the cuvette in the apparatus ensures that temperature changes in the cuvette rapidly and accurately reflect those applied, as verified (*Figure 2—figure supplement 2*). Importantly, the mito-gTEMP fluorescence ratio, and its calibration against externally applied temperature, provides a second check on the findings obtained using MTY. Although the principle of the mito-gTEMP method is different, being based on fluorescent reporter proteins rather than an externally applied dye, the calibration steps rely on the same, validated assumptions, and produced temperature readouts very similar to those derived using MTY, although typically several °C lower in the case of each of the mammalian cell lines tested (*Figure 3E*). As stated earlier, the MTY-based measurements may be slightly in error (by ~2°C), due to the minor deviation from linearity of the MTY fluorescence temperature-response curve, in the inferred temperature range of 50–60°C (*Figure 2—figure supplement 3B*).

Although the two methods gave essentially concordant results, it is important to recognise that both have limitations. Apart from our failure to obtain a stable fluorescence reading for S2 cells treated with oligomycin, we did not experience problems working with MTY on any of the cell lines tested in the presence of single OXPHOS inhibitors, such as were reported earlier for a human primary fibroblast line (*Chrétien et al., 2020*). However, using combinations of OXPHOS inhibitors, other than rotenone plus antimycin, we were not able to infer mitochondrial temperature by the MTY method, due to unstable fluorescence readings (*Figure 2—figure supplement 5A*). These are likely due to dye leakage resulting from membrane-potential anomalies of the type reported earlier (*Chrétien et al., 2020*). In particular, we note that most combinations of OXPHOS inhibitors that failed to give a stable MTY fluorescence reading involved oligomycin plus an inhibitor of respiration. Since we already know that a complete loss of membrane potential leads to leakage of the dye, we surmise that collapse of membrane potential is the most likely reason for the fluorescence instability. In the presence of oligomycin alone, a minimal respiratory electron flow balanced against proton leakage should suffice to maintain a membrane potential. Similarly, even when respiration is inhibited, ATP synthase alone should be able to generate a membrane potential. However, the membrane potential may collapse when both oligomycin and a respiratory chain inhibitor are simultaneously applied.

Various confounding factors could potentially influence MTY fluorescence. In previous studies it was argued that drugs that have different effects on mitochondrial membrane potential nevertheless all lead to a sustained increase in MTY fluorescence. Since it is not possible to stain cells simultaneously with MTY and membrane-potential sensitive dyes such as TMRM, due to spectral overlap, we need to rely on experiments conducted under parallel conditions, both by ourselves (*Cannino et al., 2012*) and others (*Kalbácová et al., 2003*; *Yang et al., 2021*). Treatment of cells with respiratory inhibitors such as rotenone or antimycin lead to substantially decreased membrane potential (*Cannino et al., 2012*; *Kalbácová et al., 2003*), whereas the effects of oligomycin on mitochondrial membrane potential are generally minor, and vary between cell lines and over time. Commonly, oligomycin results in a small increase in mitochondrial membrane potential, sometimes followed by a decrease (*Kalbácová et al., 2003*; *Yang et al., 2021*). In contrast, we here observed a sustained increased of MTY fluorescence in all of the mammalian cells treated with any of these inhibitors. This implies that any quantitative effect of membrane potential on MTY fluorescence is negligible, unless the potential is completely lost, leading to dye leakage. Any confounding effect upon MTY fluorescence of other physiologically varying parameters, such as calcium concentration, hydrogen peroxide, or pH, was directly excluded (*Figure 2—figure supplement 4*).

The excitation wavelength for gTEMP (360 nm) falls within the ultraviolet range, and the resulting cellular damage under continuous illumination for more than 60 min renders calibration unreliable (*Lu et al., 2022*). An alternative probe set, based on the recently developed B-gTEMP (*Lu et al., 2022*), should overcome the problem in future studies. In the current report, we used mito-gTEMP only up to 35 min, which generated a reliable calibration curve (*Figure 3D*). However, this is not long enough to profile very slow temperature changes such as the rewarming we observed using MTY on AOX-expressing cells (*Figure 6*). In addition, we found that some inhibitors, notably antimycin, though not rotenone or oligomycin (*Figure 3—figure supplement 2*), affected the mito-gTEMP fluorescence ratio, compromising its use. Similarly, the effects of BKA (*Figure 5—figure supplement 1*), nPG (*Figure 6—figure supplement 2B*), and MitoBloCK-6 (*Figure 2—figure supplement 5B*)

on MTY fluorescence readings preclude their use with the probe. One important caveat regarding the application of mito-gTEMP is that it was necessary to extrapolate the calibration curve beyond 42°C, assuming the linearity seen earlier for the isolated proteins (*Nakano et al., 2017*). Above this temperature, in vivo calibration is not possible, due to the loss of cellular viability and integrity, accompanied by major changes in protein composition and distribution (*Roti Roti, 2008*).

The finding that mitochondrial temperature is of the order of 50°C or more and distinct from that of the rest of the cell has been challenged on both theoretical (*Baffou et al., 2014*) and experimental grounds (*Moreno-Loshuertos et al., 2023*). However, recent data indicate that the slow thermal diffusivity of mammalian cells (*Lu et al., 2022*) is sufficient to account for the maintenance of intracellular temperature differences inferred under physiological conditions and further documented here. Furthermore, the maintenance of high mitochondrial temperature as a source of intracellular heat is supported by theoretical considerations (*Kang, 2018*).

It should also be noted that previous studies using different probes have implied that the temperature across the cell is not uniform. Some organelles or active regions manifest distinct temperature inhomogeneities: notably, the nucleus is 1–2°C warmer than the cytoplasm (*Okabe et al., 2012*; *Hayashi et al., 2015*), the nucleolus even hotter (*Piñol et al., 2020*), and perinuclear mitochondria may be hotter than those at the cellular periphery (*Savchuk et al., 2019*).

It will therefore be of great interest to use the temperature reporters employed here to track how the various treatments and stresses that we imposed affect not only mitochondria, but all of the other organelles and cellular components with which they interact. A simple way this could be done systematically would be to reclone the gTEMP reporters (or better, the improved B-gTEMP version) for targeting to other compartments, followed by careful validation of targeting, e.g., to nucleus, ER lumen, mitochondria-associated membranes, peroxisomes, endosomes, Golgi, the plasma membrane, and cytosol. Such a study would provide a picture of how alterations in mitochondrial metabolism and heat production affect intracellular temperature and its regulation more globally. An alternative approach would be to employ the recently developed palette of small-molecule intracellular temperature reporters, the Thermo Greens (*Liu et al., 2022*). Earlier studies using temperature-sensitive dyes targeted specifically to the ER (*Arai et al., 2014*; *Itoh et al., 2016*; *Kriszt et al., 2017*) have indicated how the temperature of this organelle responds to external stresses and physiological stimuli. Use of the B-gTEMP reporters targeted to different sub-mitochondrial compartments would also shed light on heat transfer within mitochondria, and how this varies with organelle architecture in different cell types. The use of reporters in isolated mitochondria might also provide insight into how much buffering of heat transfer is provided by the cytosol, although this would not address the temperature of the organelle itself in vivo. Note, however, that not all such dyes and reporters may be employed simultaneously, although studies in parallel should help address many of the outstanding questions. Since every method has technical limitations, it is obviously desirable that all conclusions arising from our and similar studies should, as far as is feasible, be independently verified using different approaches.

We previously reported (*Chrétien et al., 2018*) that isolated OXPHOS complexes I and V are heat labile, though OXPHOS complex II (cII), cIII, and cIV activities have temperature optima in the 50–55°C range when studied in intact mitochondrial membranes. Others have recently reported that the imposition of an external temperature of 43°C or above to cells or to isolated mitochondria damages respiratory function and disturbs OXPHOS complex and supercomplex organisation (*Moreno-Loshuertos et al., 2023*). Thus, externally applied heat delivered to mitochondria, which our assays indicate are already at elevated temperatures, may push them beyond their operational limit, implying that mitochondrial heat production in vivo must be finely attuned to cellular needs, to avoid catastrophic overheating. On the other hand, a recent meltome analysis of human proteins found that those constituting the respiratory chain complexes have median Tm values of around 60°C (*Jarzab et al., 2020*), consistent with our inference of high mitochondrial temperature.

## Inhibition of each OXPHOS complex produces specific temperature effects on mitochondria

As the OXPHOS system is traversed from cI through cV, we observed significant increases in the temperature effect of specific inhibitors, with essentially the same pattern observed in two iMEF lines, U2OS, HEK293T, and even in *Drosophila* S2 cells, as far as methods allowed, i.e., oligomycin>KCN>antimycin≧rotenone. Oligomycin inhibits all protonmotive components of the OXPHOS

system, including those that feed electrons via ubiquinol to cIII, even though they are not directly protonmotive. The lesser amount of mitochondrial temperature decrease produced by blocks on cI, cIII, or cIV (*Figure 2*) is not the result of incomplete inhibition: the inhibitor doses were pre-checked on all cell lines by respirometry and are similar to those used in many other studies. Instead, we posit that it reflects the proportion of electron flow entering the respiratory chain at each step. Since rotenone and antimycin inhibition had the same mitochondrial temperature effect in iMEFs, both individually and in combination, we infer that electron flow entering the chain independently of cI in this cell background is negligible. In contrast, antimycin produced a significantly greater mitochondrial temperature decrease in U2OS cells than did rotenone (*Figure 2B*), from which we conclude that around 20–30% of electron flow into cIII in this cell line must come from other ubiquinone-linked enzymes. In iMEFs (*Figure 2C and D*) and S2 cells (*Figure 2E*), though not in HEK293T or U2OS cells, KCN had a significantly greater mitochondrial temperature effect than antimycin, which implies that, in these cell lines, a measurable amount of electron flow to oxygen enters the OXPHOS system via cytochrome *c* independently of cIII, i.e., via the Mia40-Evr1/ALR protein import pathway, albeit that we were not able to verify this directly using the Evr1/ALR inhibitor MitoBloCK-6. There may also be other, as yet uncharacterised, electron donors to cytochrome *c*. In all cell lines that we were able to test, oligomycin produced a significantly greater mitochondrial temperature decrease than KCN (*Figure 2A–D*). Since KCN should block all respiratory proton pumping across the IMM, we infer that cV itself must be able to generate some heat in the complete absence of cellular respiration, presumably by the hydrolysis of mitochondrially imported ATP. Since this residual thermogenesis is inhibited by oligomycin, it must nevertheless involve cV, in a reaction that is itself protonmotive, and hence limited by the rate of dissipation of the IMM proton gradient by other processes, such as protein, metabolite, and ion transport.

In cells expressing AOX, the initial effects of OXPHOS inhibition at these various steps was different (*Figure 6A and B*). Rotenone inhibition of electron flow through cI initially produced the same temperature decrease in AOX-expressing iMEF mitochondria as in control iMEF mitochondria (*Figure 6A and B*). This is consistent with the fact that AOX by-passes cIII+cIV but not cI (*Dassa et al., 2009*; *McDonald and Gospodaryov, 2019*). In contrast, the mitochondrial temperature decrease produced by antimycin inhibition of cIII was substantially less than in control cells (*Figure 6B*). Electron flow through the inherently thermogenic AOX is expected substantially to replaces the normal heating effect of electron flow through cIII+cIV. Note that, when cIII and cIV are able to function normally, electrons flowing through cI should not be diverted via AOX (*Szibor et al., 2020*). However, the mitochondrial temperature decrease brought about by antimycin in AOX-expressing MEFs was not zero. Part of the explanation is that the capacity of AOX to support electron flow from cI through to oxygen in these cells remains below that of cIII+cIV, as inferred previously by respirometry (*Giordano et al., 2019*). Furthermore, the full engagement of AOX for cI-linked substrates (*Szibor et al., 2020*) requires the accumulation of ubiquinol, the reduced form of coenzyme Q. Nevertheless, the initial temperature decrease produced by antimycin in the mitochondria of AOX-expressing cells is maintained only for a relatively short time, approximately 10 min (*Figure 6C*), before rewarming commences. The fact that the initial effect of KCN in AOX-expressing cells was greater than that caused by antimycin supports the idea that some of the electron flow through cIV in iMEFs is introduced via cytochrome *c* independently of cIII, for which AOX does not provide a by-pass.

## High mitochondrial temperature is maintained under diverse metabolic stress conditions

Despite the fact that, under the rather extreme conditions of toxic inhibition of the OXPHOS complexes, high mitochondrial temperature is compromised, we uncovered several contexts wherein it appeared to be homeostatically maintained, hinting at one or more metabolic adaptation processes. First, it appears to decline only slowly and minimally in response to nutrient starvation, and to be restored by renewed substrate availability (*Figure 5*), consistent with previous estimates by an independent method, using the so-called upconversion nanoparticles (*Di et al., 2021*). This implies the existence of a metabolic reserve even in cells such as fibroblasts that are not normally considered repositories for storage molecules such as triglycerides or glycogen. Second, in cells expressing AOX we recorded a gradual restoration of high mitochondrial temperature after the initial decrease provoked by inhibition of the OXPHOS complexes (*Figure 6A and C*). Although AOX is not normally present in vertebrate cells, the metabolic flexibility that it confers may be important in maintaining mitochondrial

temperature and organismal viability in the vast array of species that do possess AOX but are subject to wide external temperature fluctuations, namely bacteria, fungi, plants, eukaryotic protists, and many animals as well (**McDonald and Gospodaryov, 2019**). The exact nature of the metabolic changes that enable AOX-endowed cells to recover normal mitochondrial temperature over periods of 40–70 min remains to be elucidated and may be complex. Logically it should involve a remodelling of metabolism towards the oxidation of flavin-linked substrates that directly reduce ubiquinone. Rewarming was evident in the case of cells in which cI was inhibited by rotenone (**Figure 6A and C**), indicting that it does not involve the acquisition of additional capacity for AOX to receive electrons via cI. Rather, it must involve a metabolic switch to other pathways, even though via AOX these do not support ATP production, only heat generation and redox homeostasis.

The rather minimal effect of BKA, i.e., a possible transient, small increase in mitochondrial temperature, followed by an equally modest decrease, was also surprising. BKA treatment of immortalised fibroblasts has previously been reported to induce a switch to glycolytic ATP production and decreased flux through the respiratory chain, whilst maintaining high mitochondrial membrane potential (**Kano et al., 2019**). The modest mitochondrial temperature decrease that was inferred may simply correspond with that seen in cells deprived of substrate over the time period of observation (compare **Figure 5A and E**, or **Figure 5B and F**). Thus, the efficient exchange of adenine nucleotides between mitochondria and the cytosol is not required for the maintenance of high mitochondrial temperature. Instead, the usually minor contribution of intramitochondrial ATP turnover may be boosted sufficiently to drive mitochondrial heat production to near-normal levels, possibly involving, once again, a contribution from cV-linked ATP hydrolysis.

The most surprising result that we obtained was the maintenance of high mitochondrial temperature in cells subjected to prolonged treatment with anisomycin at a dose in excess of that which profoundly inhibits cytosolic protein synthesis (**Grollman, 1967**; **Sidhu and Omiecinski, 1998**). Protein synthesis uses between 35% and 70% of the ATP produced in cells (**Stouthamer, 1973**; **Pontes et al., 2015**; **Siems et al., 1992**; **Rolfe and Brown, 1997**). We therefore expected that the predicted decrease in ATP demand would lead to decreased flux through the respiratory chain and to a sustained drop in mitochondrial temperature. Instead, anisomycin treatment led to a modest but transient increase (~4°C) in mitochondrial temperature (**Figure 7A**), with gradual reversion to the original mitochondrial temperature after 2–3 hr in the presence of the drug. Even after prolonged (18 hr) anisomycin treatment, leading to widespread cell death, high mitochondrial temperature was maintained in the surviving cells. The same effects, but without cell death, were also seen in AOX-expressing cells (**Figure 7B**).

Similarly, overnight growth in media that are predicted to stimulate mitochondrial as opposed to cytosolic ATP production had no effect on mitochondrial temperature (**Figure 2—figure supplement 5C**). Once again, the precise metabolic changes that underlie these phenomena remain uncertain and may not be easy to dissect. Nevertheless, our observations imply a homeostatic mechanism adjusting mitochondrial heat production to metabolic supply and demand, keeping mitochondrial temperature within a narrow range.

Based on the inhibitor studies shown in **Figure 7C and D**, mitochondrial temperature maintenance in anisomycin-exposed iMEF cells is lower and less dependent on cI with time, yet remains fully dependent on cV. This suggests that, as cellular ATP usage dwindles, cV may start to operate thermogenically as an ATPase, although this does not explain how the resulting proton gradient across the IMM is dissipated. A more complex mechanism, such as the futile creatine cycle recently documented in brown fat (**Rahbani et al., 2021**), may need to be invoked. The adenine nucleotide carrier (ATP-Mg/Pi), encoded by a small multigene family, which ensures the net delivery of adenine nucleotides to the mitochondrial matrix under specific physiological conditions (**Fiermonte et al., 2004**), may also be involved in sustaining a futile cycle of intramitochondrial ATP hydrolysis, as well as possible reverse electron transport through cI. In anisomycin-exposed, AOX-expressing cells, where AOX itself provides an alternative mechanism of thermogenesis, cells remained more dependent on cI for the maintenance of high mitochondrial temperature over time (**Figure 7C**), but less dependent on cV (**Figure 7D**).

An important caveat of our findings is that we cannot formally distinguish whether metabolic adjustments are being made in response to external stresses, with mitochondrial temperature maintenance just a 'by-product' of these changes or, conversely, that mitochondrial temperature is directly

responsive to external stresses, entraining metabolic adjustments in its wake. A search for an endogenous mitochondrial temperature sensor and an associated signal transduction machinery may help resolve this.

## Theoretical considerations

A mitochondrial temperature in excess of 50°C places severe constraints on the mechanisms of intracellular heat conductance. Here, we postulate that, at least in mammals, mitochondrial heat production sustains the cell broadly at around 37°C, but must be responsive both to external heat stresses and to metabolic changes taking place within the cell, including those within the mitochondrion itself. Thus, the molecular and physical mechanisms that govern heat flow within and out of mitochondria must be both robust and flexible.

As *Fahimi and Matta, 2022*, have pointed out, the thermal conductivity of the mitochondrial membranes must be very small at the distances encountered in the cell between the hot and cold compartments. *Fahimi and Matta, 2022* termed this the mitochondrial paradox and offered a physical explanation based on spikes in local temperature associated with proton passage from the IMS to the matrix. They posit that the dehydration and deprotonation of the hydronium ion on the IMS side and the reverse process on the matrix side creates a mode of heat transfer consistent with the time-scale of these processes and with the very low thermal conductivity of ATP synthase. Based on these considerations they were able to predict a 7K difference across the membrane. To our knowledge, their work is the only physically sound hypothesis yet proposed, to account for the surprising temperature differences inferred experimentally, and whose validity has been widely debated (*Lane, 2018*; *Baffou et al., 2014*; *Kiyonaka et al., 2015*; *Suzuki et al., 2015*; *Baffou et al., 2015*).

Matta and colleagues have also argued, based on thermodynamic principles (*Matta and Massa, 2015*), that the true efficiency of ATP synthase must be close to 90%, taking account of heat dissipation by informational transactions (Landauer's principle, *Landauer, 1961*) which is compensated by the inherent electrostatic potential of the enzyme (*Vigneau et al., 2022*). This may partly explain why the additional temperature decrease produced by inhibiting ATP synthase in addition to inhibiting respiration is both small, but non-negligible.

## Biological implications of high mitochondrial temperature

Although the amount of mitochondrial temperature decrease produced after OXPHOS inhibition by oligomycin, when measured by the MTY and mito-gTEMP methods, was similar, it was not identical, and the rather small difference was judged to be significant for at least two of the cell lines studied, U2OS osteosarcoma cells, and iMEFs. Although both probes are efficiently targeted to mitochondria, they appear to be localised in different sub-mitochondrial compartments. As derivatives of GFP, both of the mito-gTEMP reporter polypeptides linked to the COX8 targeting peptide should reside in the mitochondrial matrix (*Molina and Shirihai, 2009*; *Partikian et al., 1998*). Conversely, MTY is a rhodamine-related lipophilic dye whose uptake requires a mitochondrial membrane potential, and which is rapidly lost from mitochondria when the membrane potential is abrogated (*Chrétien et al., 2018*). Its localisation at or very close to the inner face of the IMM is supported by our sub-mitochondrial fractionation experiments, showing that it co-fractionates preferentially with SMPs rather than with the mitochondrial matrix-enriched fraction. This, despite reports that at least some of it interacts with the matrix protein ALDH2 (*Arai et al., 2015*). Since the IMM is also the location of mitochondrial heat generation, it should logically be slightly warmer than the matrix that it surrounds, accounting for the small temperature differential implied by the two methods.

Since high mitochondrial temperature was here documented in five different cell lines from three species of animal, including a poikilotherm, it is clearly not just an exceptional feature of one cell type. Rather it would appear to be a common property of mitochondria and of the chemistry of the OXPHOS system. Moreover, it suggests a novel role for the universal double membrane system of mitochondria, as an insulating layer enabling mitochondria to maintain a high internal temperature, whilst allowing the remainder of the cell to operate at a lower temperature. If this is correct, a corollary is that the ratio of mitochondrial volume to surface area may change in response to metabolic needs and physiological signals that regulate mitochondrial heat output. Put more simply, mitochondria might be expected to fragment into smaller entities when heat output rises, so as to maintain internal mitochondrial temperature, and to fuse into filamentous structures when heat output is low. They may

also fragment if mitochondrial heat production is simply abolished and mitochondria need to absorb as much heat as possible from the rest of the cell, in order to maintain a minimal metabolism.

This proposition could account for some of the apparently paradoxical dynamics of mitochondria which, when subject to various stresses can either undergo widespread fission or hyperfusion (*Das and Chakrabarti, 2020*). The mitochondrial fusion/fission cycle has hitherto been associated primarily with mitochondrial quality control (*Gottlieb et al., 2021*), which is disturbed in patholological states and during aging (*Seo et al., 2010*). However, mitochondrial temperature homeostasis could also be a determinant. Accordingly, it has recently been reported that inhibition of cV, but not of the other OXPHOS complexes, triggers mitochondrial fragmentation (*van der Stel et al., 2023*). Treatment with rotenone or antimycin were reported to result in rapid depletion of intramitochondrial ATP (*van der Stel et al., 2023*), consistent with our inference that these inhibitors allow some residual mitochondrial heat output driven by ATP hydrolysis, whilst oligomycin does not. Using mito-gTEMP, it was previously shown that treatment of cells with an uncoupler, which is known to provoke mitochondrial fragmentation (*Legros et al., 2002*), also resulted in a mitochondrial temperature increase of ~6°C (see Figure 4 of *Nakano et al., 2017*). Using a different spectrophotometric method, other authors estimated the effect of a chemical uncoupler in the 4–5°C range (*Piñol et al., 2020*). Consistent with the idea that mitochondrial fusion is associated with low mitochondrial heat output, the cell line with the greatest inferred temperature difference between the two methods used in the present study to assess mitochondrial temperature, and also the one where this difference was the most variable experimentally, was U2OS (*Figure 3E*), a cell line in which mitochondria are mostly filamentous, e.g., see *Westrate et al., 2014*. In other cell lines, such as iMEFs, mitochondria are highly fragmented under standard culture conditions, e.g., see *Abdullah et al., 2022*.

Staining of cells with MTY revealed bright puncta within mitochondria (*Figure 4B*), the nature of which is unknown. One possibility is that these may correspond with nucleoids. Their bright staining could indicate that they are cooler than their mitochondrial surroundings, or could be an artefact, due to preferential binding of the dye to one or more components of the nucleoid. In *Saccharomyces cerevisiae* the orthologue of ALDH2 (Ald4) is a nucleoid protein (*Kucej and Butow, 2007*), but this has not been reported to be the case in mammals.

Since the OXPHOS system is of bacterial origin, its heat output in bacteria should be similar to that of mitochondria, and we would expect that the thermal insulation provided in gram-negatives by the double membrane and lipopolysaccharide (LPS) cell wall, or by the thicker cell wall in gram-positives, should be at least as effective as that of the double membrane of mitochondria. It will therefore be interesting to implement thermosensitive reporters such as gTEMP or B-gTEMP in bacteria, to assess intracellular temperature, which we would predict to be once again some 15–20°C warmer than ambient temperature. This has important potential implications for understanding the metabolic biochemistry of bacteria, including antibiotic action.

Similarly, considering that heat generation is an inescapable cellular function of metabolically active mitochondria, a loss of this activity, as in cases of mitochondrial disease, may compromise the maintenance of intracellular temperature or induce futile cycles of cytosolic ATP generation and usage that may have profound metabolic effects. This could be a crucial aspect of mitochondrial pathology (*Rango et al., 2014*) and may also influence progression in other metabolic diseases, including cancer (*El-Gammal et al., 2022*). Some of these and related issues are discussed in the recent paper of *Fahimi et al., 2020*.

## Conclusion

On the basis of our findings, we propose that the maintenance of high mitochondrial temperature should be regarded as an important homeostatic process of the organelle, along with more traditionally considered parameters such as membrane potential, pH, ATP, metabolite levels, and ROS.

## Materials and methods
### Cell culture

All cell lines used are freely available and are not subject to any ethical restrictions on use. Human embryonic kidney-derived cells expressing a mutant version of the SV40 large T antigen, HEK293T (*Lebkowski et al., 1985*, originally obtained from ECACC), U2OS (2T) osteosarcoma cells (*Pontén*

*and Saksela, 1967*) (originally obtained from ECACC), two isolates of immortalised mouse embryonic fibroblasts generated in-house and characterised previously (*Andjelković et al., 2018*; *Giordano et al., 2019*), iMEF(L7) (*Giordano et al., 2019*) and iMEF(P) (*Andjelković et al., 2018*), and immortalised mouse embryonic fibroblasts expressing *C. intestinalis* alternative oxidase, iMEF(AOX) (*Andjelković et al., 2018*), also generated in-house and characterised previously (*Andjelković et al., 2018*) were cultured in DMEM medium containing 4.5 g/L glucose, 10% heat-inactivated FBS, and 100 U/mL each penicillin and streptomycin, at 37°C in 5% $CO_2$ with 95% humidity. Cells were also grown in low-glucose medium (Sigma, D5546) containing 1 g/L glucose, supplemented with 25 mM galactose where indicated. *Drosophila melanogaster* S2 (Schneider 2) cells (Invitrogen), originally derived from a primary culture of late stage (20–24 hr of age) embryos (*Schneider, 1972*), were cultured in suspension in Schneider's medium (Sigma S9895-1L) supplemented with 10% heat-inactivated FBS, 100 U/mL each penicillin and streptomycin at 25°C and diluted 1:6 every 3–4 days. Cell lines were authenticated in each case by the supplier and tested regularly for mycoplasm.

## Transfection of mammalian cells with mito-gTEMP

To avoid the use of neomycin/geneticin selection, HEK293T cells already being resistant, the mito-gTEMP ratiometric reporter pair (see *Figure 3A*) was recloned from mito-gTEMP_pcDNA3 (Addgene, plasmid #109117) into the pTriEx-1.1 Hygro plasmid (Novagen 70928). Stably expressing cell lines were generated using FuGENE HD Transfection Reagent (Promega), according to the manufacturer's instructions. Based on preliminary trials, hygromycin selection was implemented at the following doses: iMEF(P) and U2OS – 145 µg/mL, iMEF(AOX) 240 µg/mL, and HEK293T 450 µg/mL.

## Spectrofluorometry

For spectrofluorometric measurement of mitochondrial temperature using MTY (*Arai et al., 2015*), sub-confluent cells were treated with 100 nM MTY supplemented with warmed fresh medium, for 15 min. Cells ($5×10^6$ HEK293T and U2OS cells, $7×10^6$ iMEFs, and $10×10^6$ S2 cells) were trypsinised and collected by centrifugation at $250 × g_{max}$ for 3 min. The pellet was washed once in 10 mL PBS warmed to 37°C, then maintained as a concentrated pellet at 37°C for 10 min, for anaerobiosis. Cells were resuspended in PBS in a magnetically stirred 3.5 mL quartz cuvette with 10 mm optical path (Hellma Analytics, Germany) and placed in a Peltier temperature-controlled chamber set at 38°C (25°C for S2 cells). The time-based fluorescence signal at constant emission (excitation 542 nm, emission 562 nm) was generated using a QuantaMaster-6/2003 LPS-220B spectrofluorometer (PTI-Horiba, Japan, *Figure 2—figure supplement 1*). For ratiometric spectrofluorometry of cell lines genetically expressing mito-gTEMP, cell preparation and collection were carried out similarly, with the emission of both fluorophores and their ratio recorded over time, with excitation at 360 nm and emission for Sirius at 425 nm and for mT-Sapphire at 509 nm.

## Ratiometric temperature calibration for mito-gTEMP in vivo

Cells were prepared as described above and, once resuspended in the cuvette, were treated with 5 µM oligomycin to block ATP synthesis, hydrolysis, and respiratory electron transfer. As soon as the fluorescence ratio reached a steady state, the temperature of the Peltier jacket holding the cuvette was shifted to 30°C and then in 3°C steps to 42°C, in each case recording the fluorescence ratio during the third minute. The experiment was repeated (n=9) and the ratios of mT-Sapphire/Sirius fluorescence were plotted against temperature to generate a linear calibration function.

## Sub-mitochondrial fractionation

iMEFs were labelled with MTY as above, on 2×20 cm plates, with unlabelled cells treated in parallel. Following trypsinisation, cells were washed with 10 mL of ice-cold PBS and mitochondria isolated by homogenisation under hypotonic conditions, as described (*Lampl et al., 2015*). Mitochondria were resuspended in 2 mL of hypotonic buffer (5 mM HEPES, 2 mM EDTA, 1 mM PMSF, pH 7.2) plus 5 mM each of pyruvate, glutamate, and malate (PGM) mix. Following incubation for 5 min on ice, digitonin was added to 1% (vol/vol), with gentle shaking on ice for 15 min, followed by centrifugation at $20,000 × g_{max}$ for 10 min at 4°C (Centrifuge 5810R, Eppendorf) (*Beutner et al., 2005*; *Pallotti and Lenaz, 2007*; *O-Uchi et al., 2013*) to separate mitoplasts (pellet) from outer mitochondrial membrane and IMS components (supernatant). For estimating fluorescence, the pellet was resuspended in the

hypotonic buffer plus PGM to the same volume (2 mL) as the supernatant, and both samples were brought to 38°C before fluorescence measurement (excitation 542 nm, emission 562 nm), which was carried out over 1 min for all samples to verify that it had reached a constant value. In a further fractionation step, mitoplasts were sonicated on ice at high power, using five 30 s on/off cycles, i.e., for 5 min (Bioruptor Sonicator, Diagenode) (*Beutner et al., 2005*; *Pallotti and Lenaz, 2007*; *O-Uchi et al., 2013*), then centrifuged at 27,000 × $g_{max}$ for 15 min at 4°C (Optima XPN-100 Ultracentrifuge, Beckman Coulter, with SW60 Ti rotor). Resuspension of the pellet (SMPs) in the same volume as the supernatant (matrix fraction), temperature equilibration at 38°C, and fluorescence measurements were performed as above. The autofluorescence of control cell samples was subtracted from the fluorescence of the corresponding fraction of MTY-labelled cells, enabling the proportionate recovery of label in different fractions to be estimated. See *Figure 3—figure supplement 3* for a summary of the fractionation scheme.

## Respirometry

In order to determine the exact concentrations of inhibitors required for complete inhibition of each OXPHOS complex, we used the same amount of cells as used for spectrofluorometry to measure oxygen consumption with a Clark-type electrode (Oxytherm System, Hansatech, UK). Intact cell respiration was recorded from cells suspended in 500 μL of Respiraton Buffer A (225 mM sucrose, 75 mM mannitol, 10 mM Tris/HCl, 10 mM KCl, 10 mM $KH_2PO_4$, 5 mM $MgCl_2$, pH 7.4) plus 10 mg/mL freshly added BSA, permeabilised by the addition of 80 μg/mL digitonin at 37°C. Substrate concentrations were as follows: 10 mM ADP, 5 mM pyruvate and malate (cI+cIII), 10 mM succinate (cII+cIII), 50 μM TMPD/1 mM ascorbate (cIV). The required concentrations for full inhibition were determined to be 3 μM rotenone (cI inhibitor), 1 μM antimycin (cIII inhibitor), 0.8 mM potassium cyanide (KCN, cIV inhibitor), 5 μM oligomycin (cV inhibitor), and 100 μM nPG (AOX inhibitor).

## Thermo-profiling of mitochondria in cells treated with OXPHOS inhibitors

Cells were prepared for spectrofluorometry, and inhibitors were added to the above concentrations, once fluorescence or fluorescence ratio had reached a steady state, generally 10 min after initial oxygenation. Fluorescence values were then studied over time, until a new equilibrium value was reached, but only until 35 min for mito-gTEMP. MTY fluorescence was calibrated against temperature in each experiment, by raising and lowering the Peltier by 3°C at the end of the experiment, and recording the fluorescence values reached. The change in temperature was then estimated using this internal calibration. For estimating temperature changes using mito-gTEMP, the calibration function described above was applied to the actual values obtained for the mT-Sapphire/Sirius ratio. The ANT inhibitor BKA (Sigma, B-6179) and ALR/Evr1 inhibitor MitoBloCK-6 (Calbiochem, Sigma-5.05759) were used similarly at 100 μM.

## Cytoplasmic protein synthesis inhibition

Cells were seeded and, after 24 hr, the medium was replaced with otherwise complete medium without penicillin or streptomycin, but with 150 μM anisomycin (A-9789, Sigma). Cells were incubated for various times up to 18 hr. For measuring the mito-gTEMP fluorescence ratio, cells were then treated as for control (untreated) cells expressing mito-gTEMP, except that for all time-points except that at t=0, the cells were resuspended in PBS containing 150 μM anisomycin. For measuring MTY fluorescence in anisomycin-treated cells, 100 nM MTY was added 15 min prior to each time-point, followed by resuspension in PBS containing 150 μM anisomycin and the addition of OXPHOS inhibitors as for control (untreated cells).

## Microscopy

Cells were seeded on glass bottom dishes (35 mm MatTek, USA) and grown for 24 hr in standard growth media at 37°C, 5% $CO_2$. The culture medium was replaced with prewarmed medium containing fluorescent dyes, namely 100 nM MTY and/or 100 nM MitoTracker Deep Red FM (Thermo Fisher Scientific). After 15 min incubation, the staining medium was removed, and cells washed once with fresh, prewarmed PBS then kept in PBS at 37°C until analyses were complete. The expression and localisation of mT-Sapphire in transgenic cell lines was imaged by spinning disc (X-light

V2, CrestOptics) confocal fluorescence microscopy (Nikon FN1), with excitation at 360 nm (Cool-Led pE-400 light source) and emission filter 525/50 nm, with Nikon 60×/1.0 dip objective (CFI Apochromat NIR 60× W). Laser excitation light intensity was adjusted to minimise photobleaching. Fluorescence images for MitoTracker Deep Red FM (excitation 633 nm, emission 700/75 nm) were generated similarly. Live-cell and time-lapse images of cells stained with MitoTracker Deep Red FM or MTY (excitation 542 nm, emission 562 nm) were captured by laser scanning microscopy (Nikon A1R), using a Nikon 60×/1.27 water-immersion (CFI SR Plan Apo IR 60XC WI) objective. The detector sensitivity was adjusted for each sample to optimise the image brightness and to avoid saturation. The Nikon A1R with N-SIM laser scanning confocal microscopy system includes a live-cell supporting chamber where $CO_2$, temperature of the culture dish (surrounded by heated-water supply), and temperature of objective in use can be controlled. Time-lapse imaging was performed on cells maintained at 37°C with 1 image/10 s for 10 min, with a 2 min pause for addition of oligomycin to 5 μM, followed by a further 30 min of image capture. Signal analyses of time-lapse images were performed with ImageJ (Fuji), by surface-plot analysis.

## Statistics

All data are presented as means ± SD. Statistical significance was calculated by standard unpaired one-way ANOVA with Tukey HSD test (https://www.astatsa.com) or, where appropriate, by Student's t test (LibreOffice Calc). Values of $p < 0.05$ were considered significant. For full details and exact p values, see source data files.

## Image presentation

Micrographic images and fluorescence traces were cropped, rescaled, and optimised for brightness and contrast, with labels added or removed, but with no other modifications.

## Materials availability statement

Plasmids encoding mito-gTEMP for expression in mammalian cells (*Figure 3A and B*) are freely available on request. Cell lines available from public repositories or commercially are as described in Materials and methods. Immortalised mouse embryo fibroblasts from control and AOX-expressing mice, as first described in *Andjelković et al., 2018*; *Giordano et al., 2019*, are available upon request.

## Acknowledgements

We thank Malgorzata Rak and Eric Dufour for useful discussions, Kateryna Gaertner for assistance with subcellular fractionation, Tea Tuomela for technical assistance, Maria Carretero-Junquera for artwork and the Biocenter Finland-supported Tampere Imaging Facility for access to its infrastructure. Funding: This work was supported by Academy of Finland (award 324730 to HTJ and TSS, awards 322732 and 328969 to TSS), Basic Science Research Institute Fund (2021R1A6A1A10042944) and National Research Foundation of Korea (NRF) grant funded by the Korea government (MSIT) (2023R1A2C300453411), both awarded to Y-TC, the Sigrid Juselius Foundation (grant 3122800849 to TSS) and by a grant from Core Research for Evolutionary Science and Technology, Japan Science and Technology Agency (JPMJCR15N3 to TN).

## Additional information

### Funding

| Funder | Grant reference number | Author |
|---|---|---|
| Academy of Finland | 324730 | Tiina S Salminen Howard T Jacobs |
| Basic Science Research Institute Fund | 2021R1A6A1A10042944 | Young-Tae Chang |
| Sigrid Juséliuksen Säätiö | 3122800849 | Tiina S Salminen |

| Funder | Grant reference number | Author |
|---|---|---|
| Japan Science and Technology Agency | 10.52926/jpmjcr15n3 | Takeharu Nagai |
| National Research Foundation of Korea | 2023R1A2C300453411 | Young-Tae Chang |
| Academy of Finland | 322732 | Tiina S Salminen |
| Academy of Finland | 328969 | Tiina S Salminen |

The funders had no role in study design, data collection and interpretation, or the decision to submit the work for publication.

## Author contributions

Mügen Terzioglu, Conceptualization, Resources, Data curation, Formal analysis, Supervision, Validation, Investigation, Visualization, Methodology, Writing – original draft, Writing – review and editing; Kristo Veeroja, Conceptualization, Validation, Investigation, Methodology, Writing – review and editing; Toni Montonen, Resources, Investigation, Visualization, Methodology, Writing – review and editing; Teemu O Ihalainen, Conceptualization, Resources, Supervision, Visualization, Methodology, Writing – review and editing; Tiina S Salminen, Conceptualization, Resources, Supervision, Funding acquisition, Project administration, Writing – review and editing; Paule Bénit, Conceptualization, Methodology, Writing – review and editing; Pierre Rustin, Conceptualization, Resources, Methodology, Writing – review and editing; Young-Tae Chang, Takeharu Nagai, Conceptualization, Resources, Funding acquisition, Methodology, Project administration, Writing – review and editing; Howard T Jacobs, Conceptualization, Resources, Formal analysis, Supervision, Funding acquisition, Validation, Visualization, Methodology, Writing – original draft, Project administration

## Author ORCIDs
Tiina S Salminen ⓘ http://orcid.org/0000-0002-7232-0754
Pierre Rustin ⓘ http://orcid.org/0000-0002-5500-7280
Howard T Jacobs ⓘ https://orcid.org/0000-0003-1895-6003

Reviewer #1 (Public Review): https://doi.org/10.7554/eLife.89232.3.sa1
Reviewer #2 (Public Review): https://doi.org/10.7554/eLife.89232.3.sa2
Reviewer #3 (Public Review): https://doi.org/10.7554/eLife.89232.3.sa3
Author Response https://doi.org/10.7554/eLife.89232.3.sa4

# Additional files

## Supplementary files
• MDAR checklist

## Data availability
All experimental data used in the paper are presented in the accompanying source data files.

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
