## [Editor Report · eLife assessment]

The study provides **useful** data supporting prior findings that mitochondria in cultured cells maintain a temperature up to 15°C above the external temperature at which cultured cells are maintained. The evidence supporting the hypothesis is **solid**, although direct measures of temperature in isolated mitochondria or comparison with other cellular compartments would have strengthened the ability to interpret the relevance of the findings. Nevertheless, the bioenergetic implications of the work will be of interest to cell biologists, biochemists, and physiologists.

---

## [Referee Report · Reviewer #1 (Public Review)]

Terzioglu and co-workers tested the provocative hypothesis that mitochondria maintain an internal temperature considerably higher than cytosolic/external environmental temperature due to the inherent thermodynamic inefficiency of mitochondrial oxidative phosphorylation. As a follow-up to a prior paper from some of the same authors, the goal of this study was to conduct additional experiments to assess mitochondrial temperature in cultured cells. Consistent with the prior work, the authors provide consistent evidence that the temperature of mitochondria in four different types of cultured mammalian cells, as well as cells from *Drosophila* (poikilotherms), is 15oC or more above the external temperature at which cells are maintained (e.g., 37oC). Additional evidence shows that mitochondria maintain higher temperatures under several different types of cellular metabolic stresses predicted to decrease the dependence on OxPhos, adding to the notion that natural thermodynamic inefficiency and heat generation may be an important, and potentially regulated, characteristic of mitochondrial metabolism.

Strengths

Demonstration that both a fluorescent (Mito Thermo Yellow) and a genetic-based (mito-gTEMP) mitochondrial targeted temperature probe elicit similar quantitative changes in mitochondrial temperature under different experimental conditions is a strength. The addition of the genetic probe to the current study supports prior findings using the fluorescent probe and thus achieves a primary objective of the study.

The experiments are well designed and executed. Specific attention given to potential artifacts affecting probe signal and/or non-specific effects from the different experimental interventions is a strength.

The use of different cultured cell lines from different organisms provides additional evidence of elevated temperature as a general property of functioning mitochondria, representing additional validation.

Weakness:

While the findings and potential interpretations put forward by the authors are intriguing, the severity of the interventions (e.g., mitochondrial complex-specific inhibitors, inhibition of protein synthesis) and the absence of simultaneous or parallel measurements of other key bioenergetic parameters (i.e., membrane potential, oxygen consumption rate, etc.) limits the ability to interpret potential cause and effect - whether the thermogenesis aspect of OxPhos is being sensed and regulated, or whether temperature changes are more of a biproduct of adjustments in OxPhos flux under the experimental circumstances. In other words, the physiological relevance of the findings remains unclear.

Related, several of the interventions are employed to either increase or decrease dependence on OxPhos flux, but no outcome measures are reported to document whether the intended objective was achieved (e.g., increased OxPhos flux in low glucose plus galactose, decreased ATP demand-OxPhos flux with anisomycin, etc.).

---

## [Referee Report · Reviewer #2 (Public Review)]

An important paper that confirms the validity of the initial findings of Chretien et al regarding the hot temperatures at which the mitochondrion is operating. The authors responded adequately to the reviewers' concerns.

---

## [Referee Report · Reviewer #3 (Public Review)]

The goal of this study was to use a combination of fluorescent dyes and genetically encoded reporters to estimate the temperature of mitochondria. The authors provide additional evidence that they claim to support "hot" mitochondria.

Strengths:

1. The authors use several methods, including a mitochondrial fluorescent reporter dye, as well as a genetically encoded gTEMP temperature probe, to estimate mitochondrial temperature.

2. The authors couple these measurements with other perturbation of mitochondria, such as OXPHOS inhibitors, to show consistency

Weaknesses:

1. The methodology for inferring mitochondrial temperature is not well-established to begin with and requires additional controls for interpretation.

a. Very little benchmarking is done of the "basal" fluorescence ratio, and whether that fluorescence ratio actually reflects true organelle temperature. For instance, the authors should in parallel compare between different organelles to see if only mitochondria appear "hot" or whether this is some calibration error. Another control is to use different incubator temperatures and see how mitochondrial (vs other organelle) temperature varies as a function of external temperature.

b. The authors do not rigorously control for other factors that may also be changing fluorescence and may be confounders to the delta fluorescence (eg, delta calcium in response to mito inhibitors, membrane potential, redox status, ROS, etc.). There should be additional calibration for all potential confounders.

c. Can these probes be used in isolated mitochondria and other isolated organelles. Such data would also help to clarify whether the high temperature is specific to mitochondria.

2. The authors should try to calibrate their fluorescence inference of temperature with an alternative method and benchmark to others in the field. For instance, Okabe et al Nat Comm 2012 used a polymeric thermometer to measure temperature and reported 33degC cytoplasm and 35degC nucleus. Can the authors also show a ~2degC difference in their hands between those two compartments, and under those conditions are mitochondria still 10degC hotter?

Based on the aforementioned weaknesses, in my opinion, the authors did not achieve their Aims to accurately determine the temperature of mitochondria. The results, while interesting, are preliminary and require additional controls before conclusions can be drawn. Previous studies have indicated intra-organelle temperature variations within cells; typically, previous reports have estimated that the variation is within a few degrees (Okabe et al Nat Comm 2012). Only one report has previously suggested that mitochondria are at 50degC (Cretien, Plos biology 2018). The study does not substantially clarify the true temperature of mitochondria or resolve potential discrepancies in previous estimates of mitochondrial temperature.

---

## [Author Response]

The following is the authors’ response to the original reviews.

Response to Editor and Reviewers

Terzioglu et al, Mitochondrial temperature homeostasis resists external metabolic stresses

**Editor:**

We greatly appreciate the specific direction of the editors in guiding us as to what experiments areneeded to strengthen the manuscript for publication. We here summarize how we have handled thisadvice (please refer to response to specific reviewer points, below, for the details). Changes to the text are indicated by red text and marginal red boxes numbered as per the responses below.

Benchmarking: we now include a direct calibration of MTY against temperature. Performingexperiments on temperature probes localized to different subcellular and submitochondrialcompartments would be interesting and potentially informative, but is a whole new study that wouldrequire a great deal of validation. Hopefully it will be implemented, but it would not change the basicconclusions from the current study.

Probe localization: In addition to referring to previously published literature, and the existing Figures3B, 4 and S4 indicating that both MTY and mito-gTEMP are localized in mitochondria (the latter in the matrix), we have conducted some simple experiments to determine the intramitochondrial localization of MTY, applying standard subfractionation protocols. The findings confirm our previous assumption that MTY is inner membrane-associated.

Expected outcomes: Since, in most cases, it is not possible to do this simultaneously with fluorescence measurements, we rely mostly on previous literature which is fully cited, or on measurements conducted in parallel (e.g. respirometry, Fig. S5) or previously in our own laboratories (e.g. flow cytometry on TMRM-stained cells). We accept that specific inferences on causality, e.g. that the effect of anisomycin is mediated by decreased ATP usage, or that the effects of Gal medium are to enforce dependence on OXPHOS, are arguably an over-reach. We have therefore toned down these statements so as to focus on the mt temperature response to the treatments, rather than to the imputed downstream physiological effects thereof.

Confounding factors: We tested (and excluded) possible confounding factors affecting MTY and report the findings in an expanded supplementary figure.

Discussion of the model(s) proposed by Matta: We have now included this, as far as we consideredappropriate for the eLife readership. However, not being theoretical physicists, we would greatlywelcome a careful scrutiny of what we have written, by both the reviewer and handling editor.

**Reviewer #1:**

A1. Causality: We agree with the reviewer in that we cannot formally distinguish, in this study, whether metabolism is adjusted to maintain mitochondrial temperature, or whether mitochondrial temperature maintenance is a secondary consequence of metabolic changes induced by stress. We have added a note to the Discussion to this effect. On balance, we would argue that the many cases that we have documented here tend to favour the former assertion, although this does not constitute proof. Identification of a sensor of mitochondrial temperature changes and an associated signal transduction machinery to orchestrate responses to it would be needed to settle this, but we are obviously very far from this at present. We have added this point to the Discussion, as well.

A2. Metabolic correlates: We concede that the reviewer has a valid point, although exploring itsramifications in detail is not straightforward. The effects of AOX on respiration and resistance toOXPHOS inhibitors are documented previously and are also included in the paper as a check (Fig. S5). Our starting assumptions were that cells grown in low glucose/galactose would depend more upon mitochondrial as opposed to glycolytic ATP production, whilst net ATP production in anisomycin-treated cells should be attenuated, due to decreased ATP demand. Nevertheless, there are a number of ways this could be achieved, especially if our suggestion that altered ATP production is balanced by decreased or increased futile ATP turnover geared to maintenance of mitochondrial temperature. For example, measuring total oxygen consumption, P to O ratio or steady-state levels of ATP (or any other metabolite) would not be definitive. To accommodate the reviewer’s point, we have made clear that the various treatments we applied are predicted to alter metabolism in the specified ways, based upon theoretical arguments and previous data. To establish the exact details of the metabolic changes that accompany these treatments would require tracer-based metabolomics over time (see Jang 2018, 10.1016/j.cell.2018.03.055), followed up by measurements of specified enzyme activities. Whilst this would be very useful data that may illuminate our observations, it is obviously beyond the scope of the present paper. We hope that future studies will eventually unravel the relationship between metabolic adaptation and mitochondrial temperature.

A3. Combinations of inhibitors: We were (and remain) reluctant to cram the paper too full ofunsubstantiated speculations. Most, though not all, of the combinations of OXPHOS inhibitors thatfailed to give a stable reading of MTY fluorescence involved oligomycin plus an inhibitor ofrespiration. Since we already know that a complete loss of membrane potential leads to leakage of the dye, we surmise that this is the most likely reason for the fluorescence instability. In the presence of oligomycin alone, the minimal respiratory electron flow sustained should suffice to maintain amembrane potential if balanced against proton leakage. Conversely, even when respiration is inhibited, ATP synthase alone should be able to generate a membrane potential. However, the membrane potential may collapse when both oligomycin and a respiratory chain inhibitor are simultaneously applied. We expanded our comment on this issue in the Discussion and referred to it, briefly, in the legend of Fig. S3A.

A4. Figure 4A: We added the panel indicators to the figure.

A5. Fig.7C: We have tried to tighten up the wording, for clarity. Yes, the blue trace was the relevantdata, but we were comparing the effect of rotenone on cells treated with anisomycin for 1, 2….18 hours with cells not treated with anisomycin at all (i.e. blue trace, zero h time-point).

A6. Meaning of ‘control iMEFS’ (Fig. 7C): We meant iMEFs not expressing AOX. We have made thestatement more precise, accordingly.

A7. Supplementary Movie S1: The movie was sent, to accompany the submission. If it is not accessible for review, please contact the handling editor.

**Reviewer #2:**

B1. Theoretical considerations (‘mitochondrial paradox’): Since we are not theoretical physicists, we have deferred to the reviewer’s expertise in these matters and quoted the suggested literature assuccinctly as possible for the largely biological audience of eLife, sticking closely to the reviewer’sown words. In this light, we would invite the reviewer to scrutinize our added text (in a short additional section of the Discussion, for both this and point B3, below), and suggest any rewording that they consider appropriate.

B2. Biological implications: We appreciate the point, but since the Discussion section is already long,we have just referred the reader to the treatment of Fahimi et al. We hope to expand on these issues in a separate paper, to be published elsewhere.

B3. Theoretical considerations (Landauer’s principle and ATP synthase electrostatics): Once again, we have mentioned the issue as suggested, but would ask the reviewer to check the exact language we have used and propose any amendments they consider necessary.

**Reviewer #3:**

C1. Benchmark comparisons: We acknowledge that there are limitations to the use of each method of mitochondrial temperature assessment, and we now explain them more thoroughly in a new section of the Discussion. However, the fact that the two methods give approximately the same result constitutes a crucial validation. In addition, we verified the temperature-responsiveness of MTY fluorescence in free solution at physiological pH (see new supplementary figure panel, Fig. S2D), showing that the response is almost linear over the temperature range inferred in the experiments (35-65 ºC). Note, however, that the response curve generated cannot be used directly for calibration, due to the unknown contributions in vivo from cellular autofluorescence and quenching under OXPHOS-inhibited conditions, which may modify the signal, and will vary according to the amount of dye taken up in a given experiment. Because of this, the internal calibration used in each experiment is a far more reliable way of relating observed fluorescence changes to temperature. Note, however, that if the slight deviation from linearity seen at higher temperatures in the MTY fluorescence temperature-response curve (dotted line in Fig. S2D) reflects how the dye responds in vivo, MTY-based estimations of mitochondrial temperature may be over-estimated by ~2 ºC. This is now made clear in the text.

C2. Basal temperature: The basal mitochondrial temperature (no inhibitors) as inferred from the mitogTEMP calibration curve was already in the paper zero time points for iMEF(P) and iMEF(AOX)cells, Fig. 7A, 7B.

C3. Other organelles: In principle, gTEMP could be targeted to other organelles, such as the nucleus, peroxisomes, ER, plasma membrane and so on, which would be highly informative in profiling intracellular temperature heterogeneities. However, this would require further rounds of recloning and expression, followed in each case by verification of intracellular targeting; obviously quite a large study beyond the scope of our present work. In any case, it would now best be undertaken using the improved, next-generation ratiometric probes (B-gTEMP), which is under way. We agree that this is an important question for future experimentation and have added a short extra section to the Discussion, accordingly.

C4. Variation with external temperature: We implemented additional experiments to test this, subjecting cells to a mild heat- or cold-shock, and tracking MTY fluorescence both before and after the subsequent addition of oligomycin, with final internal calibration as before. The results were again qualitatively reproducible, but suggested that the combination of external temperature shock and bioenergetic stress. We show the details of the results of these experiments here, for the reviewer and others to inspect and consider. However, since they are not straightforwardly interpretable, we feel that they should be reserved for a future study which investigates the effects of external temperature changes on intramitochondrial temperature and bioenergetics in much greater detail. For these reasons we show the data here only, and not in the revised paper.

Both cold shock (38→32 ºC) and heat shock (38→41 ºC) produced immediate shifts of mt temperature, but by lesser amounts than the external stresses applied, i.e. a cooling of 2-4 ºC in the first case and a warming of 0-2 ºC in the second. Over the following 10 min the mt temperature of the temperature-shocked cells held steady or drifted only slightly. These observations are broadly consistent with the general conclusions of the paper that mitochondrial temperature resists external stresses. However, the effect of then adding oligomycin was intriguingly different from that seen in control cells. In cold-shocked cells the mt temperature shift produced by oligomycin was several degrees less than in control cells and mitochondrial temperature then gradually readjusted upwards to near the starting value, suggesting the induction of thermogenic pathways to compensate for the decreased external temperature. In heat-shocked cells, the response to oligomycin was reproducibly triphasic: the initial cooling effect was less pronounced than in control cells, but was followed by rewarming and then by a prolonged and progressive cooling. This is obviously much harder to interpret, and will require substantial further studies to parse.

C5. Other factors: Although this point is addressed in previous literature, we measured effects directly in solution (for MTY). Note, however, that it is not feasible to measure membrane potentialsimultaneously, due to the spectral overlap between e.g. TMRM and MTY. Nevertheless we were able to test the effects on MTY fluorescence of incremental changes in Ca2+, pH and ROS within the physiological range (see doi: 10.1073/pnas.95.12.6803, doi: 10.1074/jbc.M610491200 and doi:10.3390/antiox10050731). The results clearly indicate that changes in any of these parameters has no effect on MTY fluorescence (new supplementary figure panels S3E, S3F and S3G).

C6. Localization of probes: The existing Figures 3B, 4 and S4, as well as previous literature, indicate a mitochondrial localization both for MTY and mito-gTEMP. The matrix localization of proteins of theGFP reporter family tagged with the COX8 matrix-directed targeting signal used here is wellestablished (e.g. see doi: 10.1016/S0076-6879(09)05016-2). To investigate the sub-mitochondriallocalization of MTY we conducted a standard series of fractionation steps, using detergents,centrifugation and sonication. Whilst these do not provide absolute purity, they clearly indicate thatMTY in energized mitochondria resides in or closely associated with the inner mitochondrialmembrane. In two trials, in which mitochondria were fractionated into mitoplasts versus outermembrane/inter-membrane space fractions, an average 92% of the MTY fluorescence was retained in the mitoplast fraction (after subtracting autofluorescence from control samples not treated with MTY). After sonication, which should render most of the inner membrane pelletable as ‘inside out’ submitochondrial particles (SMPs), leaving most of the matrix contents in solution, 90% of the MTYfluorescence signal (again based on two trials, with background subtracted) was recovered in the SMP fraction, supporting the proposition that the dye is inner-membrane associated. These findings are now reported in the Results section and commented on in the appropriate section of the Discussion. We agree with the reviewer that it would be useful to target temperature probes, e.g. B-gTEMP, to specific sub- and extra-mitochondrial compartments (cytosol, MAMs, outer membrane, IMS, inner membrane or even specific protein complexes therein), so as to gauge the nature of intramitochondrial heat conduction between compartments and its radiation to the extramitochondrial environment. However, because it would be an extensive study in its own right, requiring careful validation of targeting, we feel this should be attempted as a follow-up study.

C7. Use of probes in isolated mitochondria: In principle we see no reason why this should not work, but any result would be non-physiological, since the external environment of isolated mitochondria is not the complex protein- and organelle-rich environment of the cytoplasm, which must play a crucial role in modulating heat diffusion from the organelle. Such an experiment may be useful to assess how much temperature buffering is provided by the rest of the cytoplasm, even though it does not directly address the internal temperature of mitochondria in vivo. Accordingly, we added a sentence to the Discussion foreshadowing such an experiment.

C8. Other probes and methods: See points C1 and C3 above. The reviewer’s suggestion could best be addressed using the superior B-gTEMP reporters engineered for specific expression in the nucleus and cytosol. This would be part of an extensive new study beyond the scope of the present work, but would of course be a further validation of its conclusions. We agree that multiple approaches are needed to address the issue of temperature differences within cells, in light of the surprising findings both of ourselves and of others, such as the study of Okabe et al (2012) to which the reviewer refers. This point too is now added to the Discussion.

C9. Theoretical considerations: The critiques referred to are now briefly addressed in the revisedDiscussion, along with those raised by Reviewer 2. However, since we are not theoretical physicists we do not feel qualified to enter the debate further. As Baffou and colleagues point out, in https://doi.org/10.1038/nmeth.3552, “In order for the community to come to a consensus, we believesome effort will be required to identify the actual origin of the signal measured in these studies, boththeoretically and experimentally“. Our experimental findings provide source data for this debate but do not resolve it.